# Comparing the Performance of Julia on CPUs versus GPUs and Julia-MPI versus Fortran-MPI: a case study with MPAS-Ocean (Version 7.1)

Siddhartha Bishnu[1,2], Robert R. Strauss[1,3], and Mark R. Petersen[1]

[1]Computational Physics and Methods Group, Los Alamos National Laboratory, NM, 87545, USA
[2]Department of Earth, Atmospheric and Planetary Sciences, Massachusetts Institute of Technology, Cambridge, MA 02139, USA
[3]Center for Nonlinear Studies, Los Alamos National Laboratory, NM, 87545, USA

**Correspondence:** Siddhartha Bishnu (siddhartha.bishnu@gmail.com)

**Abstract.** Some programming languages are easy to develop at the cost of slow execution, while others are fast at runtime but much more difficult to write. Julia is a programming language that aims to be the best of both worlds—a development and production language at the same time. To test Julia's utility in scientific high-performance computing (HPC), we built an unstructured-mesh shallow water model in Julia and compared it against an established Fortran-MPI ocean model, MPAS-Ocean, as well as a Python shallow water code. Three versions of the Julia shallow water code were created, for: single-core CPU; graphics processing unit (GPU); and Message Passing Interface (MPI) CPU clusters. Comparing identical simulations revealed that our first version of the Julia model was 13 times faster than Python using NumPy, where both used an unthreaded single-core CPU. Further Julia optimizations, including static typing and removing implicit memory allocations, provided an additional 10–20x speed-up of the single-core CPU Julia model. The GPU-accelerated Julia code was almost identical in terms of performance to the MPI parallelized code on 64 processes, an unexpected result for such different architectures. Parallelized Julia-MPI performance was identical to Fortran-MPI MPAS-Ocean for low processor counts, and ranges from 2x faster to 2x slower for higher processor counts. Our experience is that Julia development is fast and convenient for prototyping, but that Julia requires further investment and expertise to be competitive with compiled codes. We provide advice on Julia code optimization for HPC systems.

## 1 Introduction

A major concern in computer modeling is the trade-off between execution speed and code development time. In general, programs in scripting languages like Python and Matlab are faster to develop due to their simpler syntax and more relaxed typing requirements, but are limited by slower execution time. On the other end of the spectrum, we have compiled languages like C/C++ and Fortran, which have been extensively used in scientific computing for many decades. Programs in such languages are blessed with faster execution time, but are cursed with stricter and more cumbersome syntax, leading to slower

development time. The Julia language strikes a balance between these two categories (Perkel, 2019). It is a compiled language with execution speed similar to C/C++ or Fortran, if carefully written with strict syntax (Lin and McIntosh-Smith, 2021;

Gevorkyan et al., 2019). It is also equipped with more convenient syntax and features, such as dynamic typing, to accelerate code development in prototyping. To this day, the majority of scientific computing models are programmed in compiled languages, which execute fast but can take can take years to develop—for example, the first version of MPAS-Ocean required three years (Ringler et al., 2013). In this paper, we investigate the feasibility of writing Julia codes for computational physics simulations, since a Julia program cannot only ensure high performance but also less development time in the initial stages.

We develop a shallow water solver in Julia and compare its performance to an equivalent Fortran code.

An additional complication in choosing the best language is that layers of libraries have been added to C/C++ and Fortran to accommodate evolving computer architectures. For the past 25 years, computational physics codes have largely used the Message Passing Interface (MPI) to communicate between CPUs on separate nodes that do not share memory, and OpenMP to parallelize within a node using shared-memory threads. With the advent of heterogeneous nodes containing both CPUs and

35 GPUs, scientific programmers have several new choices: writing kernels directly for GPUs in CUDA (Bleichrodt et al., 2012; Zhao et al., 2017; Xu et al., 2015); adding OpenACC pragmas for the GPUs (Jiang et al., 2019); or calling libraries such as Kokkos (Trott et al., 2022) and YAKL (Norman et al., 2022) that execute code optimized for specialized architectures on the back-end, while providing a simpler front-end interface for the domain scientist. All of these require additional expertise, and add to the length and complexity of the code base. Julia also provides an MPI library for parallelization across nodes in a

40 cluster, and a CUDA library to parallelize over GPUs within a node. We have written shallow water codes in Julia that adopt each of these parallelization strategies.

In recent years, shallow water solvers have been developed in Julia by Oceananigans.jl (Ramadhan et al., 2020) and ShallowWaters.jl (Klöwer et al., 2022). These codes employ structured rectilinear meshes to discretize their spatial domain. Spearheaded by the Climate Modeling Alliance alongside independent contributors, Oceananigans has progressively matured

into an accessible and versatile software suite designed for executing finite volume simulations pertaining to incompressible fluid dynamics. This software is equipped with capabilities to operate on Graphics Processing Units (GPUs), thereby offering enhanced computational performance. It is also capable of solving both nonhydrostatic and hydrostatic Boussinesq equations. On the other hand, ShallowWaters.jl emphasizes type-flexibility and is compatible with 16-bit numerical formats in its shallow water modeling approach. This application additionally offers the benefits of mixed-precision computation and optimized

communication through reduced precision. In this paper, we conduct a comparison on unstructured-mesh models, using the Fortran code MPAS-Ocean (Ringler et al., 2013) as a point of reference. MPAS-Ocean employs unstructured near-hexagonal meshes with variable resolution capability and is parallelized with MPI for running on supercomputer clusters. We developed a Julia model employing the same spatial discretization of MPAS-Ocean, which is capable of running in serial mode on a single core, or in parallel mode on a supercomputer cluster or a graphics card. We discuss the subtle details of our implementations,

compare the speed-ups attained, and describe the strategies employed to enhance performance.

The structure of this paper is arranged as follows. Section 2 presents a comprehensive introduction to the Julia programming language, the primary subject of our experiments in this paper. This section elucidates how Julia's innovative compiler,

employing its Just-in-Time (JIT) compilation approach and dynamic type inference, equips the language with the capability to rival the performance of statically-typed languages such as Fortran and C++/C. We highlight some key features of Julia, which are not only fundamental to our research, but also provide valuable insights for researchers new to Julia, aiding their understanding of Julia's unique concepts and terminologies. In Section 3, we delineate the process of creating three versions of the Julia model: Julia-CPU, Julia-GPU and Julia-MPI. We also provide details on an equivalent Python-CPU code and the Fortran-based MPAS-Ocean, both of which serve as comparative yardsticks for assessing Julia's performance in the ensuing section. Additionally, we offer an explicit account of the hardware configurations and toolchain specifications used in the process. Section 4 furnishes the findings from our performance comparison tests, including an explanation of how we fine-tuned our preliminary model to generate the reported results. This is accompanied by an in-depth discussion and analysis of our experimental results. This section serves as a benchmark for contrasting the proficiency of Julia and Fortran in High Performance Computing (HPC) applications. In Section 5, we share insights and provide guidance to HPC developers on how to effectively utilize Julia, with an emphasis on the necessary steps to attain performance on par with that of Fortran and C++/C. This is informed by our experiences and the lessons learned throughout this experiment. Finally, Section 6 concludes the paper, encapsulating our findings and providing instructions on how to replicate our results.

## 2 Julia in a Nutshell

Julia is a high-level, just-in-time (JIT) compiled dynamic programming language, which was developed with the intention of marrying the speed of compiled languages like Fortran or C++/C and the usability of interpreted languages like Python or MATLAB. Conceived in 2012 by Shah, Edelman, Bezanson, and Karpinski at MIT (Bezanson et al., 2017), Julia has rapidly grown in popularity thanks to its innovative features and design.

As a relatively new addition to the world of programming languages, Julia benefits from the ability to incorporate the best aspects of established languages while avoiding their less convenient attributes. It provides the speed of a compiled language (owing to Just-in-Time compiling) and the simplicity of an interpreted language, making it highly appealing for developers. The language also includes a REPL (Read-Eval-Print loop) environment that interprets code lines as they are written, enhancing programmers' convenience.

Specifically designed for technical and scientific users, Julia is versatile and boasts an extensive library of mathematical functions and numerical accuracy. It also facilitates seamless interoperability with other programming languages, enabling direct calls to Fortran, C++/C, Java, or Python.

Compatible with a range of operating systems, including Windows, MacOS, and Unix, Julia has recently been gaining ground in domains requiring algebraic and numerical computing, data science and machine learning, artificial intelligence, distributed and parallel computing, and even web application development, due to its math-friendly syntax and impressive speed.

In the upcoming subsections, we succinctly delve into some of Julia's fundamental attributes, including Just-in-Time (JIT) compilation, multiple dispatch, and type hierarchy. These salient characteristics are not only pertinent to our study but will also

be instrumental in elucidating our model and interpreting the results. We then explore the intricacies of type inference in Julia, considering situations where it might falter, potentially resulting in reduced code execution speed. We wrap up this section with an explanation of Julia structs, a discourse on the diminished computational performance associated with abstract fields in these structs, and viable solutions to mitigate these challenges.

## 2.1 Just-in-Time Compilation

Julia's high performance can be attributed to one of its key features: low level virtual machine (LLVM) based on-the-fly or just-in-time (JIT) compilation, which is a combination of Ahead Of Time (AOT) compilation and interpretation. Here is a breakdown of how it works:

(a) When a function is first run in Julia, the interpreter converts the high-level code into an intermediate representation;

(b) The compiler then uses this intermediate representation to generate optimized machine code tailored to the specific types in use;

(c) This machine code is executed, and the results are returned;

(d) Most importantly, the machine code for the specific function and type combination is cached. So, if the same function is called with the same types later on, Julia can bypass the compilation step and directly execute the pre-optimized machine code.

This JIT compilation enables Julia to match the performance of statically-typed compiled languages such as Fortran, C++/C, while preserving the flexibility of dynamic languages like Python. However, it also introduces a delay referred to as 'time-to-first-plot' or compilation latency on the initial run of a function. Subsequent calls are significantly faster due to the cached machine code.

## 2.2 Function, Method, and Multiple Dispatch

In Julia, a function is a named sequence of statements that performs a computation. A method is a specific implementation of a function for particular types of arguments. A function definition starts out with a single method. But when additional definitions are provided with different combinations of argument types, the function accrues more methods. This concept is intimately tied to Julia's support for multiple dispatch, which means that the version of the function (i.e. the method) that gets called is determined by the types of all arguments. This can provide a flexible and powerful way to express program behavior.

The traditional form of multiple dispatch in Julia is dynamic or runtime dispatch. When a function is called, Julia examines the types of all arguments and chooses the most specific method that can apply to these types. The benefit of this approach is that it allows for polymorphism and code that adapts based on the types it encounters during execution.

Although not a separate dispatch mechanism, static or compile-time dispatch occurs when Julia's compiler knows the types of all arguments to a function call at compile time. In such cases, the compiler can select the appropriate method to call right

away, instead of deferring the decision to runtime. This is an optimization that can result in more efficient code, but it requires the compiler to have enough information about types, which is not always possible in dynamically-typed languages like Julia.

## 2.3   Type Hierarchy

The type hierarchy in Julia is a system that organizes all possible types into a tree-like structure, allowing for the categorization of types and subtypes.

At the top of the type hierarchy tree is the `Any` type, which is a supertype of all other types. When a variable is defined as `Any`, it can hold a value of any type. This is useful in certain scenarios where maximum flexibility is required, but it can also potentially slow down the code. This is because the compiler does not know the specific type of the values, and cannot make as many optimizations.

Abstract types serve a similar role as interfaces in other languages. They are nodes in the tree that can have subtypes but cannot be instantiated themselves. In other words, they define a kind of protocol or set of behaviors, but one cannot create objects of these types. They are only used for organizing other types into a hierarchy.

Finally, concrete types form the leaves of the tree and represent types that can actually be instantiated, but they cannot have subtypes themselves. Examples of concrete types are `Int`, `Float64`, `String` etc. Types like arrays are considered concrete only when both their element type is explicitly defined as a concrete type and their number of dimensions is specified. For example, `Array{Float64, 2}` is deemed concrete. Each concrete type is a subtype of one or more abstract types.

The type hierarchy is crucial in Julia because it enables the powerful feature of multiple dispatch, allowing functions to behave differently depending on the types of all their arguments.

## 2.4   Type Inference

In statically typed languages, such as Fortran, C++/C or Java, the programmer needs to declare the type of a variable when it is defined. This allows the compiler to generate efficient code because it knows exactly what types it is dealing with. However, Julia is designed to be easy to use like a dynamically typed language (such as Python or MATLAB), where it is not necessary to declare the types of variables. But unlike most dynamically typed languages, Julia can still produce very efficient code, thanks to its JIT compilation and aggressive type inference system, which allows the compiler to determine the type of a variable or expression without the programmer explicitly mentioning it. The compiler infers the type based on the values assigned or the operations used on the variable. However, type inference in Julia can fail or be suboptimal in a few different scenarios. Here are some of the most common ones:

(a) Functions with insufficient information about arguments: In some cases, a function might not have enough information about what arguments it will receive. This can make it hard for the compiler to infer the types.

(b) Global variables: Using global variables can lead to performance issues, because the global scope can change, which in turn prevents the compiler from inferring a stable type.

(c) Type instability: In Julia, type instability refers to a situation where the type of a variable cannot be inferred consistently by the compiler at compile time. This usually happens when the type of a variable changes within a function, or when a function's return type depends on the values (not the types) of its arguments.

(d) Containers with multiple data types: In Julia, containers are data structures used for storing collections of data. These can include arrays (an ordered collection of elements, indexed by integers), tuples (an ordered collection of elements, similar to an array, but immutable), dictionaries (an unordered collection of key-value pairs), and sets (an unordered collection of unique elements), among others. If containers are used to store different types of data, the compiler may not be able to precisely infer their types.

Even when type inference fails or is suboptimal, the Julia code should still run correctly (assuming it does not have other types of errors), but it may run slower due to the additional overhead of runtime type checking. To improve performance, it is generally a good practice to try to write type-stable code and provide the compiler with as much type information as possible.

## 2.5 Struct

In Julia, a struct, short for structure, is a composite data type, similar to a class in object-oriented languages. However, Julia's 165 struct itself is not object-oriented and has no methods directly attached to them. A struct is used to encapsulate a few related values together into a single entity, and those values are stored in fields.

Structs with abstract types or containers as fields can slow down Julia code, since they prevent the compiler from producing highly optimized, type-specific machine code. In other words, the lack of concrete type information at compile time forces the compiler to generate less efficient, generic code that can accommodate any potential subtype, resulting in performance 170 penalties from dynamic dispatch and missed optimization opportunities.

While one approach to improving the performance of Julia structs with abstract types or containers is to specify concrete types for all fields, this could reduce the flexibility of Julia's powerful type abstraction and potentially lead to repetitive code. An alternative technique involves the use of function barriers, where abstract fields are unpacked within a function that then calls an inner function, effectively passing concrete types. This strategy leverages Julia's ability to generate efficient machine 175 code based on the specific types of function arguments. However, a more elegant solution could be the use of parametric structs. With parametric structs, the type of the field is determined at the time of struct instance creation rather than at the struct definition. This approach allows the Julia compiler to generate highly optimized machine code tailored to the precise, concrete type used for each instance, significantly reducing the performance overhead typically associated with handling abstract types.

## 3 Methods

With an exploration of the relevant key features of Julia in the preceding section, we are now set to shift our focus to the details of the shallow water equations that we aim to solve, including its numerical discretization and implementation across various architectures.

## 3.1 Equation Set & TRiSK-Based Spatial Discretization

Our Julia model solves the shallow water equations (Cushman-Roisin and Beckers, 2011) in vector-invariant form. This is sufficiently close to the governing equations for ocean and atmospheric models to be used as a proxy to test performance with new codes and architectures. The equation set is

$$\boldsymbol{u}_t + qh\boldsymbol{u}^\perp = -\nabla\left(g\eta + K\right), \tag{1a}$$

$$\eta_t + \nabla \cdot (h\boldsymbol{u}) = 0, \tag{1b}$$

where $\boldsymbol{u}$ is the horizontal velocity vector, $\boldsymbol{u}^\perp = \boldsymbol{k} \times \boldsymbol{u}$, $h$ is the layer thickness, $\eta$ is the surface elevation or sea surface height (SSH), $K = |\boldsymbol{u}|^2/2$ is the kinetic energy, and $g$ is the acceleration due to gravity. If $b$ represents the topographic height and $H$ the mean depth, then $\eta = h + b - H$. Moreover, if $f$ denotes the Coriolis parameter, and $\zeta = \boldsymbol{k} \cdot \nabla \times \boldsymbol{u}$ the relative vorticity, then the absolute vorticity, $\omega_a = \zeta + f$, and the potential vorticity, $q = \omega_a/h$. The term $qh\boldsymbol{u}^\perp$ is the thickness flux of the PV in the direction perpendicular to the velocity, rotated counterclockwise on the horizontal plane. Ringler et al. (2010) refer to it as the non-linear Coriolis force since it consists of the quasi-linear Coriolis force $f\boldsymbol{u}^\perp$ and the rotational part $\zeta\boldsymbol{u}^\perp$ of the non-linear advection term $\boldsymbol{u}\cdot\nabla\boldsymbol{u}$. We spatially discretize the prognostic equations in (1) using a mimetic finite volume method based on the TRiSK scheme, which was first proposed by (Thuburn et al., 2009), and then generalized by (Ringler et al., 2010). This method was chosen to horizontally discretize the primitive equations of MPAS-Ocean while invoking the hydrostatic, incompressible, and Boussinesq approximations on a staggered C-grid. Since this horizontal discretization guarantees conservation of mass, potential vorticity, and energy, it makes MPAS-Ocean a suitable candidate to simulate mesoscale eddies.

Our spatial domain is tessellated by two meshes, a regular planar hexagonal primal mesh and a regular triangular dual mesh. Each corner of the primal mesh cell coincides with a vertex of the dual mesh cell and vice versa. A line segment connecting two primal mesh cell centers is the perpendicular bisector of a line segment connecting two dual mesh cell centers and vice versa. Regarding our prognostic variables, the scalar SSH $\eta$ is defined at the primal cell centers, and the normal velocity vector $\boldsymbol{u}_e$ is defined at the primal cell edges. The divergence of a two-dimensional vector quantity is defined at the positions of $\eta$, while the two-dimensional gradient of a scalar quantity is defined at the positions of $\boldsymbol{u}_e$ and oriented along its direction. The curl of a vector quantity is defined at the vertices of the primal cells. Finally, the tangential velocity $\boldsymbol{u}_e^\perp$ along a primal cell edge is computed diagnostically using a flux mapping operator from the primal to the dual mesh, which essentially takes a weighted average of the normal velocities on the edges of the cells sharing that edge. Interested readers may refer to Thuburn et al. (2009) and Ringler et al. (2010) for further details on the mesh specifications.

At each edge location $\boldsymbol{x}_e$, two unit vectors $\boldsymbol{n}_e$ and $\boldsymbol{t}_e$ are defined parallel to the line connecting the primal mesh cells, and in the perpendicular direction rotated counterclockwise on the horizontal plane, such that $\boldsymbol{t}_e = \boldsymbol{k} \times \boldsymbol{n}_e$. The discrete equivalent of the set of equations (1) is

$$(u_e)_t - F_e^\perp \widehat{q}_e = -\left[\nabla\left(g\eta_i + K_i\right)\right]_e, \tag{2a}$$

$$(h_i)_t = -\left[\nabla \cdot F_e\right]_i. \tag{2b}$$

where $F_e = \widehat{h_e} u_e$ and $F_e^{\perp}$ represent the thickness fluxes per unit length in the $\boldsymbol{n}_e$ and $\boldsymbol{t}_e$ directions respectively. The layer thickness $h_i$, the SSH $\eta_i$, the topographic height $b_i$, and the kinetic energy $K_i$ are defined at the centers $\boldsymbol{x}_i$ of the primary mesh cells, while the velocity $u_e$ are defined at the edge points $\boldsymbol{x}_e$. The symbol $\widehat{(.)_e}$ represents an averaging of a field from its native location to $\boldsymbol{x}_e$. The discrete momentum equation (2a) is obtained by taking the dot product of (1b) with $\boldsymbol{n}_e$, which modifies the non-linear Coriolis term to

$$\boldsymbol{n}_e \cdot \widehat{q_e} \widehat{h_e} \boldsymbol{u}^{\perp} = \widehat{q_e} \widehat{h_e} \boldsymbol{n}_e \cdot (\boldsymbol{k} \times \boldsymbol{u}) = \widehat{q_e} \widehat{h_e} \boldsymbol{u} \cdot (\boldsymbol{n}_e \times \boldsymbol{k})$$

$$= -\widehat{q_e} \widehat{h_e} \boldsymbol{u} \cdot \boldsymbol{t}_e = -\widehat{q_e} \widehat{h_e} u_e^{\perp} = -F_e^{\perp} \widehat{q_e}. \tag{3}$$

Given the numerical solution at time level $t^n = n\Delta t$, with $\Delta t$ representing the time step and $n \in \mathbb{Z}_{\geq 0}$, the Julia model first computes the time derivative or tendency terms of (2) as functions of the discrete spatial and flux-mapping operators of the TRiSK scheme. Then it advances the numerical solution to time level $t^{n+1}$ using the forward-backward method

$$u^{n+1} = u^n + \Delta t \mathcal{F}(u^n, h^n), \tag{4}$$

$$h^{n+1} = h^n + \Delta t \mathcal{G}\left(u^{n+1}, h^n\right), \tag{5}$$

where $\mathcal{F}$ and $\mathcal{G}$ represent the discrete tendencies of the normal velocity and the layer thickness in functional form, and the subscripts representing the positions of these variables have been dropped for notational simplicity.

The following sections introduce the new codes that were created for this study. Three versions of the Julia code were written (Strauss, 2023): the base single-core CPU version, an altered version for GPUs with CUDA, and a multi-node CPU implementation with Julia-MPI. These were compared against the existing Fortran-MPI and single-core Python versions of shallow-water TRiSK models. All use a standard MPAS unstructured-mesh file format that specifies the geometry and topology of the mesh, and includes index variables that relate neighboring cells, edges, and vertices. All models have an inner (fastest-moving) index for the vertical coordinate and were tested with 100 vertical layers to mimic performance in a realistic ocean model.

## 3.2 Single-Core CPU Julia Implementation

The serial-mode implementation on a single core involves looping over every cell and edge of the mesh to (a) compute the tendencies, i.e. the right-hand side terms of the prognostic equations (2), and (b) advance their values to the next time step. The tendencies can be functions of the dependent and independent variables as well as spatial derivatives of the dependent variable. The serial version of our model is the simplest one from the perspective of transforming the numerical algorithms into code.

In order to highlight differences in formulation, we provide a Julia code example for the single tendency term from (2) for the SSH gradient $-g\nabla\eta$, which is discretized as $-[g\nabla\eta_i]_e$. We then add a vertical index $k$ to mimic the performance of a multi-layer ocean model, but each layer is trivially redundant. In a full ocean model this SSH gradient term would be the pressure gradient, and would involve the computation of pressure as a function of depth and density. For the single-core CPU version, the Julia function to compute the SSH gradient is

**Listing 1.** Julia example for serial CPU

```
 1: function velocity_tendencies!(sshGradient, ssh, ...)
 2:     for iEdge in 1:nEdges
 3:         cell1 = cellsOnEdge[1,iEdge]
 4:         cell2 = cellsOnEdge[2,iEdge]
 5:         for k in 1:nVertLevels
 6:             sshGradient[k,iEdge] = - gravity / dcEdge[iEdge]
 7:                                     * (ssh[k,cell2] - ssh[k,cell1])
 8:         end
 9:     end
10: end
```

Here `cellsOnEdge` is an array of index variables describing the mesh that points to the cells neighboring an edge, and `dcEdge` represents the distance between the centers of adjacent cells sharing the edge on which the normal velocity tendency is computed. In the actual code all the tendency terms are computed within this function, but here we only show the SSH gradient as a brief sample.

## 3.3 SIMD GPU Julia Implementation

GPUs are exceedingly efficient for SIMD (Same Instruction Multiple Data) computations, leveraging their thousands of independent threads to execute the same operation simultaneously on different input values. As we are solving the same prognostic equation for (a) the SSH at every cell center $x_i$, and (b) the normal velocity at every edge $x_e$ of the mesh, a GPU naturally emerges as a powerful asset for our computations. By assigning subsets of cells and edges to distinct GPU threads, we can conduct the tendency computations and update the prognostic variables concurrently in parallel. This approach stands in stark contrast to looping over every cell and edge, an operation that would scale linearly with the size of the mesh, thereby significantly impacting the wall-clock time in large scale computations.

To harness the power of an Nvidia GPU, we crafted CUDA kernels using the CUDA.jl library (Besard et al., 2018, 2019), specifically for calculating the tendencies and updating the prognostic variables to the subsequent time step. One of the remarkable features of working with GPUs and CUDA.jl is the relative ease of code transition from a single-core to a multi-threaded GPU context. Primarily, it involved replacing the `for` loop, which iterated over cells and edges, with a more GPU-friendly design where the computation is carried out independently for each cell or edge, as shown below:

**Listing 2.** Julia example for GPU with CUDA

```
1: CUDA.@cuda blocks=cld(nEdges, 1024) threads=1024 maxregs=64
2:     velocity_tendencies_cuda!(sshGradient, ssh, ...)
3:
4: function velocity_tendencies_cuda!(sshGradient, ssh, ...)
5:     iEdge = (CUDA.blockIdx().x - 1) * CUDA.blockDim().x
6:         + CUDA.threadIdx().x
7:     cell1 = cellsOnEdge[1,iEdge]
8:     cell2 = cellsOnEdge[2,iEdge]
9:     for k in 1:nVertLevels
10:        sshGradient[k,iEdge] = - gravity / dcEdge[iEdge]
11:            * ( ssh[k,cell2] - ssh[k,cell1] )
12:    end
13: end
```

In our implementation, each cell and edge of the mesh is assigned to a distinct thread on the GPU. Thus, computations for a single cell or edge are carried out on a solitary thread, with a dedicated CUDA method enabling the mapping of thread indices to the corresponding indices of the cell ($i$) or edge ($e$) where the prognostic variable is being updated. To ensure the method's execution across all threads on the GPU, we employ a CUDA macro to invoke our kernel, specifying the number of threads to be equal to the number of cells or edges within the mesh. It is important to underscore that the core computation of sshGradient remains identical in both CPU and CUDA kernel codes, demonstrating the ease of porting computational logic from CPU to GPU context.

### 3.4 CPU/MPI Julia Implementation

Instead of cycling through each cell or edge of the mesh, we can optimize the simulation by employing multiple processors and apportioning a segment of the mesh to each one, a process referred to as domain decomposition. However, to compute certain spatial operators, we need data from the outermost cells of neighboring processors. This necessitates inter-processor communication to exchange these critical pieces of information. To streamline this communication, we introduce an additional 'halo' layer, consisting of rings of cells encircling the boundary of each processor's assigned region, which overlaps with the adjacent processors' regions. Since computation is typically much cheaper than communication, it is standard practice to perform this 'halo exchange' after at least a few time steps.

As previously mentioned, we are applying a mimetic finite volume method based on the TRiSK scheme in our calculations. Consequently, the computation of spatial operators such as the gradient, divergence, curl, and flux mapping operators (used for diagnostically computing the tangential velocities) only requires the values of the prognostic variables at the cell centers and edges of adjacent cells. Thus, to compute the spatial operators that constitute the tendencies of the prognostic variables

defined at the center and edges of a specific cell, we need to consider just one small ring of cells around the cell and the values

of the prognostic variables at the center and edges of each cell within this ring. The intersection of these small rings around

each boundary cell of the assigned region of a processor forms the innermost ring of its halo layer. When using a time-stepping

method involving $k$ tendency computations within a time step, and executing the halo exchange after $m$ time steps, the number

of rings in the halo layer is set to $n = km$. For the $q^{\text{th}}$ stage of the $p^{\text{th}}$ time step, with $1 \leq p \leq m$ and $1 \leq q \leq k$, we compute

the tendencies on the assigned region of the processor, as well as on $(m - p + 1)k - q$ rings of the halo layer, starting from the

innermost one. This process is repeated, progressively 'peeling off' the outermost ring after each tendency computation, until

after $m$ time steps, we update the values of the prognostic variables within the $mk$ rings of the halo layer via communication

with adjacent processors. In our work, we are using a forward-backward method with $k = 1$, and performing the halo exchange

after every time step, resulting in $m = p = 1$. While we acknowledge this may not be the most efficient choice, our primary

concern here is ensuring equivalent computational and communicative workload between the Fortran and Julia MPI models.

So, as long as this parity is maintained, we consider our methodology satisfactory.

Finally, implementing this parallelization approach using the mpi.jl library (Byrne et al., 2021) necessitates a few significant

modifications. For instance, we adjust the simulation methods so that each process (or rank) performs computations only for

its assigned cells or edges. We utilize the MPI communication channel (comm) to receive the updated values of the prognostic

variables in a processor's halo region from the adjacent processors that advance these variables. Conversely, we transmit the

updated values of the prognostic variables from the outermost region of the processor under consideration to the neighboring

processors, where these variables reside in the halo regions.

**Listing 3.** Julia example for CPU with MPI

```
 1: # each process executes the following, receiving a different value
 2: # on each rank:
 3: comm = MPI.COMM_WORLD
 4: rank = MPI.Comm_rank(comm)
 5:
 6: myCells = cells_for_rank(mesh_file, rank, partition_file)
 7: myEdges, myHaloEdges = edges_on_cells(myCells)
 8:
 9: velocity_tendencies!(myEdges, sshGradient, ssh, ...)
10: update_halo_edges!(sshGradient, myHalodEdges, rank, comm)
11:
12: function velocity_tendencies!(myEdges, sshGradient, ssh, ...)
13:     for iEdge in myEdges
14:         cell1 = cellsOnEdge[1,iEdge]
15:         cell2 = cellsOnEdge[2,iEdge]
16:         for k in 1:nVertLevels
```

```
17:              sshGradient[k,iEdge] = - gravity / dcEdge[iEdge]
18:                 ∗ ( ssh[k,cell2] - ssh[k,cell1] )
19:          end
20:      end
21: end
22:
23: function update_halo_edges!(data, edgesInMyHalo, rank, comm)
24:     for neighborRank in find_neighbors(rank, comm)
25:         MPI.Irecv!(data[edgesInMyHalo,:], neighborRank, 0, comm)
26:         edgesToNeighbor = find_halo_overlap(rank, neighbor, comm)
27:         MPI.Isend(data[edgesToNeighbor,:], neighborRank, 0, comm)
28:     end
29: end
```

Here `myCells` and `myEdges` are the lists of cells and edges in the local domain, owned by the rank running this code, plus its halo.

### 3.5 CPU/MPI Fortran Implementation

The baseline comparison code for this study is the Model for Prediction Across Scales (MPAS-Ocean) (Ringler et al., 2013; Petersen et al., 2015), which is written in Fortran with MPI communication commands. It is the ocean component of the Energy Exascale Earth System Model (E3SM) (Golaz et al., 2019; Petersen et al., 2019), the climate model developed by the US Department of Energy. In this study, the code is reduced from a full ocean model solving the primitive equations to simply solving for velocity and thickness (1). Thus the majority of the code is disabled, including the tracer equation, vertical advection and diffusion, the equation of state, and all parameterizations. In order to match the Julia simulations, we employ a forward-backward time-stepping scheme, exchange one-cell-wide halos after each time step, compute 100 layers in the vertical array dimension, and use the identical Cartesian hexagon-mesh domains (Petersen et al., 2022).

MPAS-Ocean is an excellent comparison case for Julia because it is a well-developed code base that uses Fortran and MPI, which have been standard for computational physics codes since the late 1990s. The highest resolution simulations in past studies used over three million horizontal mesh cells and 80 vertical layers, scale well to tens of thousands of processors (Ringler et al., 2013) and have been used for detailed climate simulations (Caldwell et al., 2019). MPAS-Ocean includes OpenMP for within-node memory access, and is currently adding OpenACC for GPU computations, but these were not used for this comparison to Julia-MPI on a CPU cluster.

### 3.6 Single-Core CPU Python Implementation

Apart from MPAS-Ocean, we examine the performance of the Julia shallow water code relative to a single-core, object-oriented Python code (Bishnu, 2022) that utilizes NumPy. This Python code employs two types of spatial discretizations to solve the rotating shallow water system of equations: the TRiSK-based mimetic finite volume method used in MPAS-Ocean, and a Discontinuous Galerkin Spectral Element Method (DGSEM). Moreover, it supports various predictor-corrector and multistep time-stepping methods, including those previously scrutinized for ocean modeling in Shchepetkin and McWilliams (2005).

The Julia shallow water code was first written by translating this Python code into Julia syntax. While the Julia code was subsequently optimized for parallelization and enhanced performance, the Python code continued to evolve as a platform to conduct a suite of shallow water test cases for the barotropic solver of ocean models. Each test case in the Python code verifies a specific subset of terms in the prognostic momentum and continuity equations, such as the linear SSH gradient term, linear constant or variable-coefficient Coriolis and bathymetry terms, and non-linear advection terms. Bishnu et al. (2022) and Bishnu (2021) offer an in-depth exploration of these test cases, discussing the numerical implementation, time evolution of the numerical error for both spatial discretizations and a subset of the time-stepping methods, and the results of convergence studies with refinement in both space and time, only in space, and only in time. However, for the purposes of the present study, only the linear coastal Kelvin wave and inertia-gravity wave test cases were implemented in the Julia code.

While this study did not leverage them, several libraries exist for enhancing Python performance across various architectures, including Numba and PyCuda for GPUs, mpi4py for CPU clusters, and Cython for single-CPU acceleration. Numba, an open-source Anaconda-sponsored project, (Lam et al., 2015) serves as a NumPy-aware optimizing JIT compiler. It translates Python functions into swift machine code at runtime, employing the robust LLVM compiler library. PyCUDA (Klöckner et al., 2012), which is structured in C++ (at its foundational layer) and Python, facilitates access to Nvidia's CUDA parallel computation API within Python. Lastly, mpi4py (Dalcín et al., 2005, 2008) offers Python bindings for the universally recognized Message Passing Interface (MPI) standard.

Another option involves 'cythonizing' an existing Python code by introducing static type declarations and class attributes, which can subsequently be converted to C++/C code and to C-Extensions for Python. Cython is an optimizing static compiler designed to yield C-like performance from Python code with supplemental C-inspired syntax. The rotating shallow water Python code (Bishnu, 2022) is currently undergoing cythonization. Once cythonized, these codes can further be accelerated on GPUs using Nvidia's HPC C++ compiler and the C++ Standard Parallelism (stdpar) for GPUs (Srinath, 2022). However, the effort required for such extensive modifications and enhancements to bring GPU-accelerated C++ algorithms to Python may not always justify the time investment. As we will illustrate in subsequent sections, a serial Julia code—already rivaling the performance of fast compiled languages—requires fewer modifications for GPU or multi-core parallelization. This makes Julia a more convenient choice for high-performance scientific computing applications compared to Python.

## 3.7 Hardware and Compiler Specifications

Multi-core CPU and GPU simulations were conducted on Perlmutter at the National Energy Research Scientific Computing Center (NERSC). In June of 2022 Perlmutter achieved 70.9 Pflop/s using 1,520 compute nodes, and was ranked 7th in the Top500 list. Erich Strohmaier (2022) Perlmutter is based on the HPE Cray Shasta platform. It is a heterogeneous system comprised of both CPU-only AMD 'Milan' nodes and GPU-accelerated 'Ampere' nodes, as detailed in Table 1. The Ampere's Nvidia A100 GPU is appropriate for this study because it is designed for HPC workloads and double precision calculations. The Julia-MPI and Fortran-MPI tests were both run with up to 64 ranks per node.

| | CPU Nodes | GPU nodes |
|---|---|---|
| **overview** | 2x AMD EPYC 7763 (Milan) CPUs | Single AMD EPYC 7763 (Milan) CPU |
| | 64 cores per CPU | 64 cores per CPU |
| | AVX2 instruction set | Four NVIDIA A100 (Ampere) GPUs |
| **memory** | 512 GB of DDR4 memory total | 256 GB of DDR4 DRAM |
| **communication** | 204.8 GB/s memory bandwidth per CPU | 204.8 GB/s CPU memory bandwidth |
| | | 40 GB of HBM per GPU with: |
| | | 1555.2 GB/s GPU memory bandwidth |
| | | 12 3rd gen NVLink links between pairs of GPUs |
| | | 25 GB/s/direction for each link |
| | PCIe 4.0 NIC-CPU connection | PCIe 4.0 NIC-CPU connection |
| | | PCIe 4.0 GPU-CPU connection |
| | 1x HPE Slingshot 11 NIC | 4 HPE Slingshot 11 NICs |
| **performance** | 39.2 GFlops per core (FP64) | 19.5 GPU TFlops (FP32) |
| | 2.51 TFlops per socket (FP64) | 9.7 GPU TFlops (FP64) |
| | 4 NUMA domains per socket (NPS=4) | 155.9 GPU TFlops (TF32, tensor) |
| | | 311.9 GPU TFlops (FP16, tensor) |
| | | 19.5 GPU TFlops (FP64, tensor) |
| **power** | 280W thermal design power per CPU | 400W thermal design power per GPU |

**Table 1.** Technical specifications for NERSC Perlmutter CPU and GPU nodes. (NERSC, 2023)

The software toolchain is as follows. Both Fortran and Julia use the MPICH implementation of the Message Passing Interface (MPI). The Fortran compiler was gnu version 11.2.0, with MPICH 3.4, which is packaged on Perlmutter with the modules `PrgEnv-gnu/8.3.3` and `cray-mpich/8.1.24`. Multi-threading was disabled (no OpenMP).

When running on a single node (up to 64 processes), we experimented with both block and cyclic distributions (run command `srun --distribution=block:block` versus `srun --distribution=cyclic:cyclic`. The block distribution would be expected to reduce communication time because it restricts processes to a single socket for 1 to 32

processes. The cyclic distribution could speed computations because processes are distributed equally across the two sockets. In practice, there was little difference between the two distributions. The figures show the block distribution. On multi-node tests, we use 64 processes per node.

The Julia version is 1.8.3 with MPICH 4.0.2. The necessary julia packages are listed in `Manifest.toml` and `Project.toml`. These packages can be installed by executing the following lines in the julia console (opened by running the julia binary with no arguments) in the root directory of the MPAS_Ocean_Julia respository:

```
]activate .
instantiate
```

Subsequently, when running julia with the flag `--project=.` in the root directory of the MPAS_Ocean_Julia repository, all the necessary packages for the environment will be loaded in their appropriate versions.

## 4 Results and Discussion

We now present the results of the verification and performance tests for our Julia model. The verification tests encompass convergence plots for the spatial operators and the numerical solution. Performance tests, on the other hand, reveal the speed-up attained by initially transforming the benchmark Python-CPU code into Julia-CPU code, performing additional optimization on the Julia-CPU code, and ultimately transitioning to the Julia-GPU code. We evaluate and compare the performance metrics of the Julia-GPU code, the Julia-MPI code, and the Fortran MPAS-Ocean code run on a single node. Lastly, we provide scaling plots comparing the Julia-MPI code with Fortran MPAS-Ocean, examining the variation of the wall-clock times with the processor count for two scenarios: strong scaling (maintaining the overall problem size constant) and weak scaling (preserving a constant problem size per processor).

For Julia-GPU and Julia-MPI computations, we sequentially measure wall-clock times for six samples, each comprising ten time steps. Although not a pragmatic approach, we execute the halo exchange for Julia-MPI (as detailed in Section 3.4) and the GPU to CPU transfer for Julia-GPU after every time step. For ten time steps, this leads to ten alternating computations and ten MPI exchanges or GPU to CPU data transfers per sample. Given the compilation latency attributable to Julia's just-in-time (JIT) compilation and caching of machine code (for subsequent use) during the initial function call (as elaborated in Section 2.1), the wall-clock time for the first sample is significantly larger, as anticipated. Consequently, we disregard the first sample as an outlier, utilizing only the succeeding five samples to compute the average wall-clock time. It is worth reiterating that in a realistic ocean model, an adequately large halo layer is designed to reduce the frequency of halo exchanges and minimize communication overhead. Similarly, when running an ocean model on a GPU, the GPU to CPU data transfers are required only when solution outputs are written to disk files.

### 4.1 Model Verification

Both serial and parallel implementations of the shallow water model, as discussed in the preceding section, were verified for accuracy through convergence tests against exact solutions. We were able to achieve the anticipated second-order convergence

of the various TRiSK-based spatial operators on a uniform planar hexagonal MPAS-Ocean mesh. These operators included gradient, divergence, curl, and the flux-mapping operator used to interpolate the tangential velocities from the normal velocities (Figure 1). These operators are formulated as shown in Figure 3 of Ringler et al. (2010). Once the operator tests were complete, the linearized shallow water equations were verified against exact solutions for the coastal Kelvin wave and inertia-gravity wave cases, as described in Bishnu et al. (2022) and Bishnu (2021). With refinement in both space and time, we observe the expected

first-order convergence of the numerical solution (Figure 1), spatially discretized with the second-order TRiSK scheme, and advanced with the first-order forward-backward time-stepping method (Bishnu, 2021).

## 4.2 Acceleration of Julia with Typing Optimizations

The first comparisons were made between the Julia serial CPU version and the reference Python CPU code, as outlined in Tables 2 and 3. The initial serial development and testing were conducted on an Intel Cascade Lake platform equipped with

an Intel Xeon processor. The performance tests detailed in this section and subsequent ones involve advancing the linear shallow water equations on a planar hexagonal mesh with 100 vertical layers. These equations incorporate the coastal Kelvin wave's exact solution to specify the initial and boundary conditions. All codes use double-precision (8 byte) real numbers, and performance tests do not include the time for initialization, input/output, or generating plots.

In its primary state, the single-core CPU Julia code, even without specific optimizations, outperformed Python by a factor

of 13. Despite both Julia and Python being dynamically typed, Julia gains a notable computational edge through its ability to infer types at runtime, perform JIT (Just-In-Time) compilation, cache and directly manipulate machine code (Section 2.1).

Following the initial development phase in Julia, further effort was put into optimization, which led to a 10–20 times speed-up for the CPU-serial code. These enhancements involved optimization for memory management by identifying and curtailing unnecessary allocations that substantially increased runtime, along with replacing the generic `Any` type with concrete types in

function definitions, and specifying or parameterizing the types of fields within structs. Detailed explanations of some of these improvements can be found in Section 5.

In order to effectively compare CPU and GPU times, one must first decide which architectures can provide a fair comparison. We chose to conduct tests on Perlmutter, with single-node CPU performance using 64 cores compared to the associated single-node GPU performance on the same machine. The results depicted in Figure 2 indicate similar wall-clock times, with

Julia-GPU times being 2–3 times slower than Julia-MPI times. Fortran-MPI speeds were comparable to Julia-GPU for larger problem sizes, but faster for smaller domains. The similarity of the full-node CPU and GPU timings is rather surprising, given the architectural differences. The listed performance for the A100 is 9.7 TFlops for 64-bit floats, while the AMD EPYC 7763 delivers 39.2 GFlops per core, resulting in a total of 2.5 TFlops for 64 cores. Based on these manufacturer specifications, we would expect the A100 to perform faster.

## 4.3 Julia-MPI versus Fortran-MPI

Julia and Fortran codes were compared on multi-node CPU clusters, where both used MPI for communication between processors. Comparisons were made with domains of 128, 256, and 512-squared grid cells solving the shallow water equations.

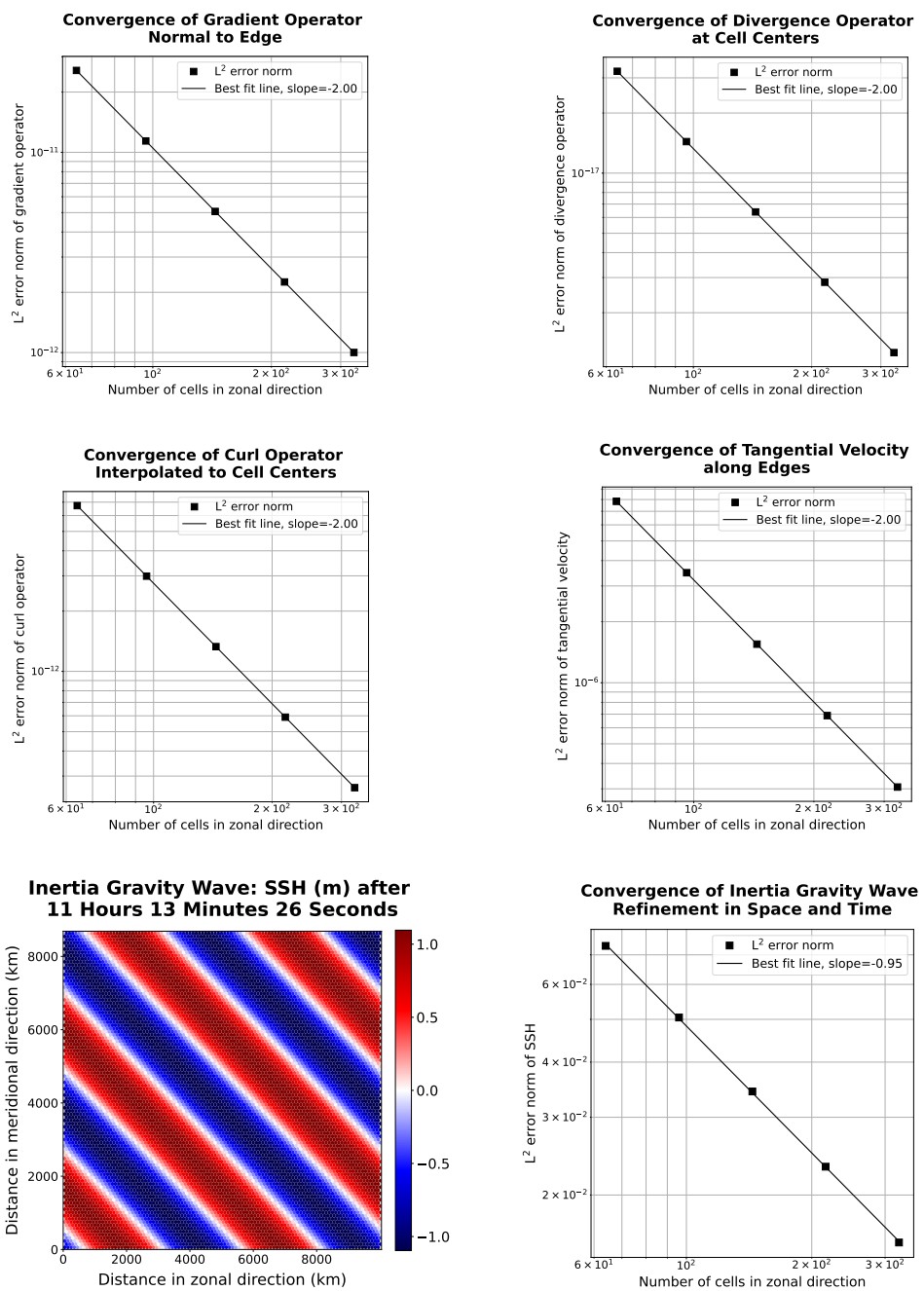

**Figure 1.** The first two rows show convergence plots of the TRiSK-based spatial operators for the newly-developed Julia code. Tests were run with both CPU and GPU implementations, and identical results were obtained. The slope of $-2$ indicates the expected second-order convergence. The third row shows a snapshot of the inertia-gravity wave test case, and the convergence plot of the numerical solution with refinement in both space and time.

|                                   | 128x128   | 256x256   | 512x512   |
| --------------------------------- | --------- | --------- | --------- |
| Python, CPU                       | 3.08E+03  | 1.31E+04  | 4.96E+04  |
| Julia, CPU-serial (unoptimized)   | 2.25E+02  | 8.64E+02  | 3.86E+03  |
| Julia, CPU-serial (optimized)     | 1.12E+01  | 7.43E+01  | 3.33E+02  |

**Table 2.** Wall clock duration (seconds) of performing ten timesteps with 100 layers on an Intel Cascade Lake CPU.

|                                   | 128x128 | 256x256 | 512x512 |
| --------------------------------- | ------- | ------- | ------- |
| Python, CPU                       | 274     | 177     | 149     |
| Julia, CPU-serial (unoptimized)   | 20      | 12      | 12      |
| Julia, CPU-serial (optimized)     | 1       | 1       | 1       |

**Table 3.** Increase in run time compared to the optimized CPU-serial Julia version at the same resolution.

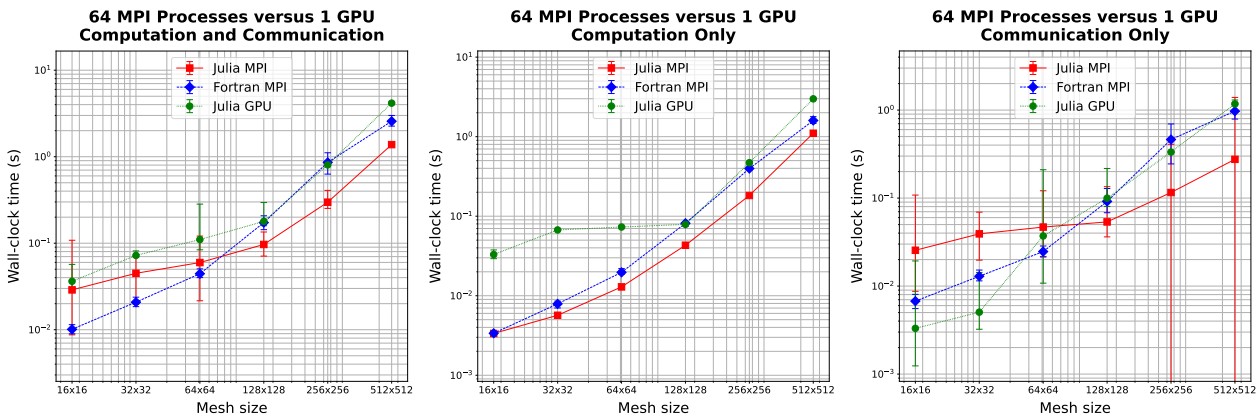

**Figure 2.** Wall clock time comparison between single-node GPU and single-node CPU (64 cores) on Perlmutter for six resolutions. Left column shows total simulation time, middle column shows time spent on computation only. Right column shows time spent on communication between MPI processes for MPI runs, or between CPU and GPU for GPU run. Error bars show the minimum and maximum collected sample.

All timing tests were conducted for 10 time steps and repeated 12 times on each processor count, spanning 2 to 2048 processors by powers of two. The vertical dimension included 100 layers to mimic ocean model arrays and provide sufficient computational work on each processor. Separate timers report on computational work versus MPI communication within the time-stepping routine. The i/o, initialization, and finalization time is excluded.

We compared the Julia and Fortran models with both strong and weak scaling. In strong scaling, the same problem size (the mesh size and number of timesteps to simulate) is run with a varying degree of parallelization. In weak scaling, the problem size scales with the degree of parallelization, such that the problem size allotted to each process is constant. In both, the duration of time it takes to complete the simulation is the dependent variable and the number of processors used for the simulation is the independent variable. This is additionally separated out into three columns: the total time to simulate the problem, the time spent on just the computation (the mathematical implementation of the equation set), and the time spent on just communicating the necessary data between processes.

The strong scaling for hexagonal meshes of 128x128, 256x256, and 512x512 cells is shown in Figure 3. In strong scaling, we expect a downwards trend of computation time with the number of processors, often giving way to a flatter behavior at high enough processor counts where communication time dominates. We indeed observe this trend with both the Julia and Fortran implementations. In the total time column, we see Julia and Fortran match very closely at lower processor counts, taking almost identical time to run. In the middle range of processor counts (16-128), Fortran takes more time than Julia. Then at high processor counts, Julia tends to become slower than Fortran, no longer scaling as well. In the computation-only column, we see that Julia is actually faster than Fortran across the board. But due to the necessary communication time (which is the dominant effect at higher process counts where insufficient work is being done by each processor) the Julia implementation is not as efficient at high processor counts as the communication time does not decrease with greater parallelization like it does for Fortran (right column.)

The weak scaling with 64, 128, and 256 cells per process is shown in Figure 4. The rows here do not represent distinct mesh resolutions like in Figure 3. Mesh size instead varies with the number of processors (resolution changes along the x-axis). In weak scaling, we expect an initially increasing trend of computation time increasing with the number of processors due to the more communication required with more processors, giving way to a flat behavior as the communication time reaches its maximum and computation is constant. Indeed, we observe a very flat behavior in the computation only column. Julia again is better across the board, while Fortran is slower in the middle range of process counts. In communication, we see Julia and Fortran increase as expected, although Fortran communication time levels out sooner while Julia is slower at communication with higher process counts, like we observed in strong scaling.

For both languages computation time scales well, decreasing at close to perfect scaling with the processor count, while communication time does not and so progressively requires a much larger fraction of time at higher processor counts (Figure 5). Once computations are optimized, communication, which is fixed by the interconnect speed, will remain a bottleneck regardless of the language (see, e.g. Koldunov et al. (2019)).

As another way of measuring scaling we keep the computational resources constant, using one node (64 processes) to compare simulation time of various mesh sizes (as shown in Figure 2). Here it is appropriate to also compare the GPU

implementation with Julia since this represents a fixed-size computational resource. We observe an increasing trend for the Julia-MPI, Fortran-MPI and Julia-GPU implementations, which we expect for an increasing problem size with constant computational power. In the middle plot, Julia-MPI and Fortran-MPI closely align, but Julia-GPU takes more time at smaller mesh sizes before eventually matching the MPI models. Launching a GPU kernel incurs some time cost. Unlike MPI, the GPU does not need to communicate data between threads, relying instead on shared memory. However, at some stage, the data generated by the computation must be transferred back to the main memory to utilize the results, whether for writing them to disk or further processing. Therefore, we measure this memory movement and contrast it with MPI time. It is vital to recognize that the frequency of this memory movement can vary significantly depending on the application. Unlike the regular MPI communication, required every few time steps to exchange information in the halo layers, this transfer might only occur infrequently. If only the final state of the model is crucial and intermediate steps don't necessitate recording, this communication time could be bypassed, potentially rendering the GPU more efficient for such scenarios.

It is notable that such a mass-parallel shared-memory based architecture as a GPU is similar to communication-based CPU-parallelization over a full node, as shown in Figure 2. For higher resolutions with sufficient work, both the compute time (middle panel) and communication time (right panel) are 2-3 times slower for Julia-GPU than Julia-MPI. Based on the technical specifications in Table 1, the computational time on the GPU would be expected to be nearly four times faster than a CPU node. The GPU has a reported performance of 9.7 TFlops for 64-bit floating point operations, while the CPU is 2.51 per CPU-core (39.2 GFlops per core, with 64 cores). Likewise, the memory bandwidth for the GPUs are nearly 8 times faster (1555.2 GB/s for the GPU versus 204.8 GB/s for the CPU). Identifying the exact reasons behind the slower-than-anticipated performance of our Julia-GPU code is challenging, but several potential factors may be at play.

(a) Shared Memory Communication with MPI: When using MPI within a single multi-core node, shared memory can be leveraged for communication, bypassing the complexity of network protocols. Unlike OpenMP, which automatically shares data between threads, MPI maintains private memory for each process. Transferring data between processes requires explicit message passing. However, shared memory communication on the same node can be substantially faster, as it reduces protocol overhead and limits memory copies. Moreover, communication within a single node typically exhibits lower latency, minimizing the time taken to initiate and complete communication events.

(b) GPU Transfer Overheads: GPU computing, on the other hand, requires every host-to-device or device-to-host transfer to proceed over the Peripheral Component Interconnect Express (PCIe) bus, introducing overheads and bandwidth limitations that can become bottlenecks, especially for frequent, small transfers. Therefore, it is plausible that frequent data transfers between MPI processes within a single node may be more efficient than equivalent transfers between a host and a GPU device. Given the same number of host-device and inter-process communications, the additional overhead of host-device transfers in the GPU code can accumulate.

(c) GPU Computation Overheads: The computational phase on the GPU could be hampered by the overhead associated with launching GPU kernels. For smaller problem sizes, this overhead becomes more pronounced, potentially underutilizing the GPU if some cores remain idle.

(d) Unstructured Mesh Memory Layout: If the code involves scattered memory accesses or neglects memory coalescing (where consecutive threads access consecutive memory locations), it can hinder efficient use of memory bandwidth, decelerating the computation significantly. Unstructured meshes inherently have a non-contiguous memory layout for neighboring elements. Such irregular and unpredictable memory access patterns can hinder the efficient use of the GPU's memory hierarchy.

(e) Optimization Challenges with Julia-GPU Code: Crafting optimal GPU code using CUDA in Julia demands a deep grasp of the GPU hardware architecture, the CUDA programming paradigm, and Julia's GPU programming capabilities. The standard methods in our code may not fully harness the GPU's potential, leaving room for further performance optimization.

In summary, a combination of these factors could be influencing the observed performance dynamics between our CPU and GPU codes.

## 5 Optimization Tips for Julia Developers

Julia serves the dual purpose of a prototyping language and a production language. It allows for the creation of code that is quick to write but slower in performance, although it is still considerably faster than other interpreted languages, as evidenced by our comparison with Python, for conceptual demonstration. Additionally, with a bit more time investment and thoughtful construction, it is possible to develop highly optimized code that achieves performance comparable to compiled languages like Fortran.

In Julia, it is the types of objects, not their values, that the compiler leverages to construct efficient machine code. This means, barring a few specific scenarios (as outlined in Section 2.4), Julia can carry out extensive type inference and generate highly optimized code without requiring explicit type declarations for variables.

Nevertheless, there are situations where type declarations can significantly enhance performance. A prime example is a struct containing fields with abstract types or containers. A more efficient approach in such cases, however, would be to transform these structs into parametric ones, a process detailed in Section 2.5. Let us consider an example in the context of our Julia-CPU and Julia-GPU codes.

In the first (unoptimized) iteration of the CPU code, we omitted specific array type declarations, allowing Julia to assign the default `Any` type:

```
struct MPAS_Ocean
    layerThickness
    normalVelocity
    ...
end
```

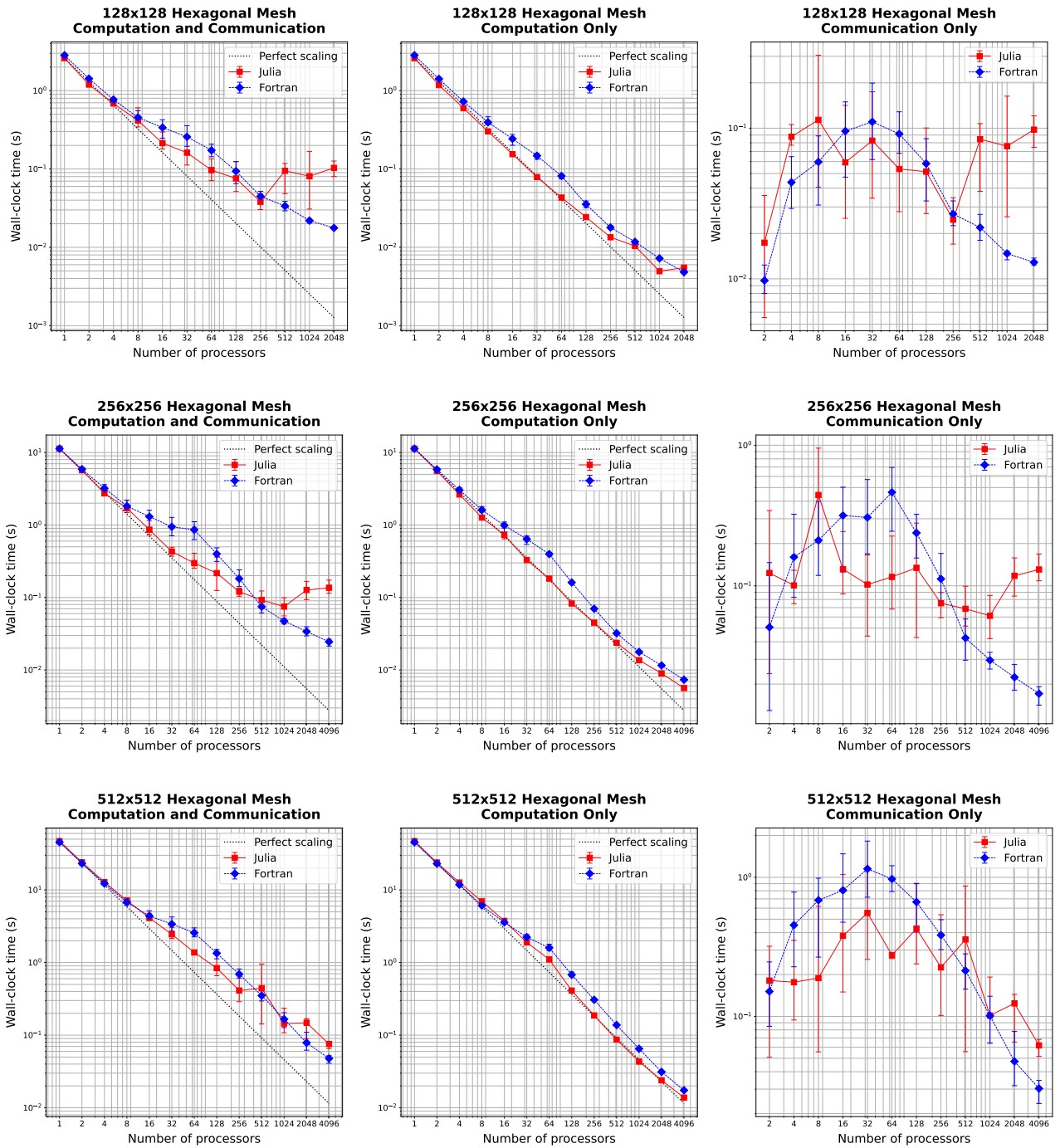

**Figure 3.** Strong scaling plots for three resolutions: $128^2$, $256^2$, and $512^2$ (rows). Left column shows total simulation time, middle column shows time spent on computation only, and right column shows time spent on communication between MPI processes. Error bars show the minimum and maximum collected sample.

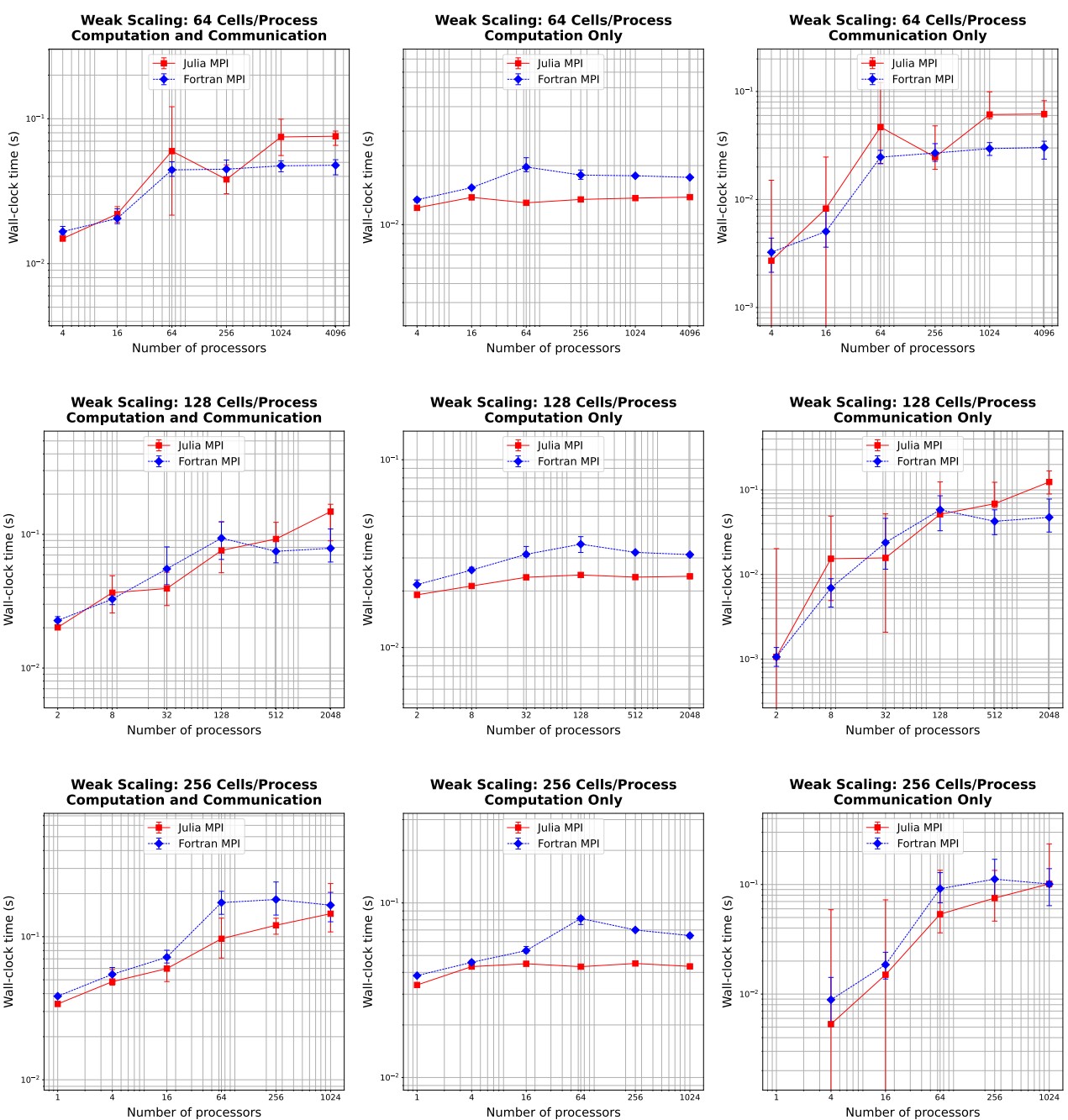

**Figure 4.** Weak scaling plots, where the problem size per processor is held fixed at 64, 128, and 256 cells/process (rows). Left column shows total simulation time, middle column shows time spent on computation only, and right column shows time spent on communication between MPI processes. Error bars show the minimum and maximum collected sample.

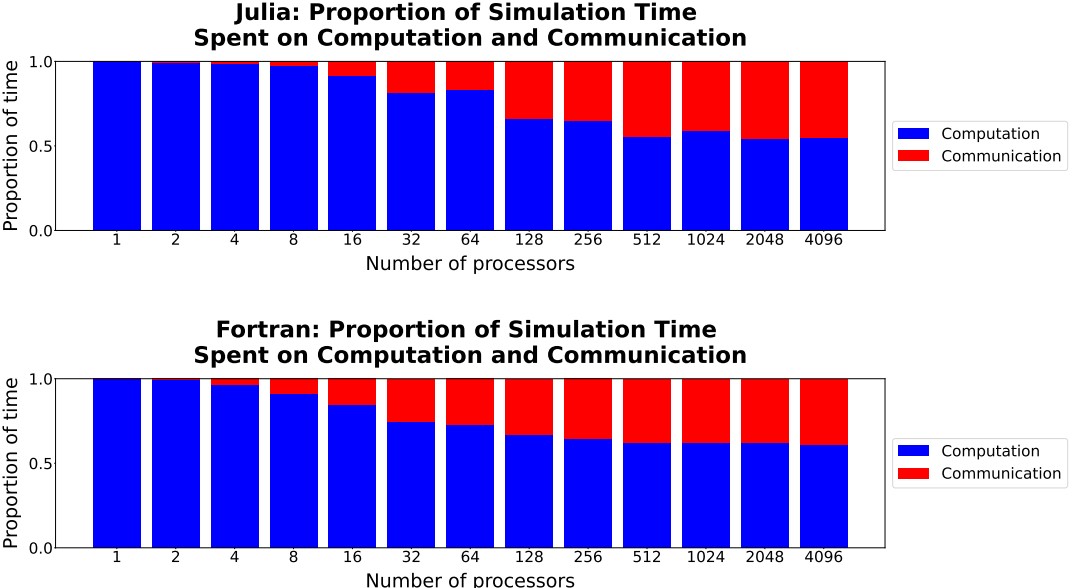

**Figure 5.** Comparison of the proportion of time spent on computation (blue) versus communication (red) in Julia-MPI (top) and Fortran-MPI (bottom) on the 512x512 hexagonal mesh with 100 layers. The relative time spent in communication increases dramatically at high processor counts since the computation time decreases with more parallelization.

By subsequently modifying these variables to be explicitly typed as two-dimensional arrays of floating-point numbers, we witnessed a substantial boost in performance:

```
struct MPAS_Ocean
          layerThickness::Array{Float64, 2}
          normalVelocity::Array{Float64, 2}
          ...
      end
```

In the process of parallelizing our code for GPU execution, we employed a different array type, `CUDA.CuArray`, specifically designed for GPU workloads. Our first approach was to create an abstract array type that could encapsulate both CPU and GPU data types. This allowed `CUDA.CuArray`s and regular `Array`s to be used interchangeably, enabling the model to operate on either the GPU or CPU as required. Additionally, we imposed a parametric constraint on the array contents (`F <: AbstractFloat`), signifying that any subtype of the abstract floating point type could be passed at runtime.

```
struct MPAS_Ocean{F <: AbstractFloat}
          layerThickness::AbstractArray{F, 2}
          normalVelocity::AbstractArray{F, 2}
          ...
```

```
    end
```

While this strategy may appear efficient because types are declared before runtime, the use of abstract types, akin to the
`Any` type, can actually hinder execution speed. At runtime, these types could be different subtypes of the abstract type,
like `CUDA.CuArray` or `Array`. This means that the specific methods to be used for these types cannot be determined at
compile-time, leading to dynamic dispatch and negatively affecting performance.

An alternative would be creating two distinct struct definitions depending on whether we are targeting GPUs or CPUs:

```
struct MPAS_Ocean_CUDA{F <: AbstractFloat}
            layerThickness::CUDA.CuArray{F, 2}
            normalVelocity::CUDA.CuArray{F, 2}
            ...
        end

        struct MPAS_Ocean{F <: AbstractFloat}
            layerThickness::Array{F, 2}
            normalVelocity::Array{F, 2}
            ...
end
```

In this approach, we are explicitly defining the types of the arrays, which can contribute to significant performance enhancement.
However, this rigid struct definition sacrifices the flexibility of dynamically switching between CPU and GPU execution, and
also requires additional lines of code for each struct definition.

A superior solution involves parameterizing both the array type and its element type within the struct definition. This
empowers the compiler to infer the concrete type of fields at compile time and optimize the code accordingly for the specific
array and element types:

```
        struct MPAS_Ocean{A<:AbstractArray{<:AbstractFloat, 2}}
            layerThickness::A
            normalVelocity::A
...
        end
```

We discovered that either of these modifications resulted in a computation speedup by a factor of 34x.

The aforementioned examples underscore the importance of parametric structs in Julia, which enable greater type flexibility
and optimize computational performance. Julia's just-in-time compiler processes functions individually, compiling each for
specific input types upon initial invocation and subsequently recompiling for novel types. Despite potential type instability
in higher-level code, type stability is achieved within the numerical kernels, the computation-intensive core of the code, as

specific types become increasingly defined. This is where parametric structs play a crucial role. By encapsulating variables of undefined types within nested function calls, they carry type uncertainty all the way down the call stack, thereby preserving type flexibility until the point of actual computation. Thus, the struct's type can adapt as necessary for the specific computational context it is used within. This functionality allows parametric structs to enhance both code flexibility and runtime efficiency, key advantages of Julia's multi-paradigm design.

A pivotal strategy in enhancing the performance of Julia code, we discovered, lies in minimizing memory allocations. Excessive memory allocation can drastically impede code execution, and it is often not readily apparent when seemingly trivial operations are culprits of unnecessary memory allocation. To illustrate, consider the act of extracting a pair of values from a two-column array:

```
cell1Index, cell2Index = cellsOnEdge[:,iEdge]
```

Surprisingly, this operation can lead to considerable memory allocation. In one particular test, this single line—recurrent throughout the simulation—was found to allocate as much as 408 KiB. This is due to the creation of a tuple, rather than a direct extraction of each column into the respective scalar variables. By dividing the operation into two distinct lines, thus bypassing tuple or array creation:

```
cell1Index = cellsOnEdge[1,iEdge]
cell2Index = cellsOnEdge[2,iEdge]
```

we successfully reduce memory allocations to zero. This modification causes the operation to be nearly instantaneous, reducing the total time spent on the entire tendency calculation by 50%, from 198 $\mu s$ to 99 $\mu s$.

One's Julia code can potentially harbor many such covert operations, contributing significantly to slower performance. Additionally, even a single struct with abstract types or containers as fields can notably hamper execution speed. Fortunately, Julia provides the @code_warntype tool for quickly identifying such memory-intensive lines:

```
@code_warntype calculate_normal_velocity_tendency!(mpas)
```

It color-codes non-concrete types and memory allocations in red, thereby directing users precisely to the lines and fields that require optimization. This singular feature elevates Julia's utility for high-performance applications, substantially accelerating the development time needed to optimize a model's performance.

Another valuable tool in the Julia optimization arsenal is --track-allocations, a command line option that can be appended to any Julia execution:

```
$ julia --track-allocations=user ./anyJuliascript.jl
```

This generates a new file at ./anyJuliascript.jl.XXX.mem (where XXX represents a unique identifier). This file presents each line of the script, prefixed by the amount of memory allocations generated by that line, providing a comprehensive line-by-line overview of where allocations occur.

## 6  Conclusions

As new programming languages and libraries become available, it is important for model developers to learn new techniques and evaluate them against their current methods. This is particularly true as computing architectures continue to evolve, and long-standing languages such as C++ and Fortran require additional libraries to remain competitive on new supercomputers.

In this work, we created three implementations of a shallow water model in Julia in order to compare ease of development and performance to standard Fortran and Python implementations. The three Julia codes were designed for single-CPU, GPU, and parallelized multi-core CPU architectures. Julia-MPI speeds were identical to Fortran-MPI at low core counts, 2x faster for mid-range, and 2x slower at higher core counts. Julia-MPI exhibited better scaling than Fortran-MPI for computation-only times, and more variability for communication times.

Julia-GPU performed very similarly to Julia-MPI, despite these implementations being based on not only vastly different architectures but entirely different parallelization paradigms (shared memory versus communication). Based on the hardware specifications, we expected the GPU version to run faster than the full-node CPU version. The lower than expected performance of the Julia-GPU code on a single node can be ascribed to several intrinsic factors, such as shared memory utilization with MPI, overheads in GPU computation and transfer, and non-contiguous memory layouts in unstructured meshes. GPU codes are particularly challenging for applications with large domains. Once simulations run on multiple nodes, the constraining factor is the inter-node (MPI) communication. This is true whether the computations within each node are mostly on the CPU or GPU. For high-resolution applications, the primary goal of GPU-enabled codes is to off-load the majority of the computations to the GPUs to take advantage of the computing power in GPU-based architectures, even if total throughput is limited by MPI communication.

The shallow water equations are simple enough for rapid development and verification, yet contain the salient features of any ocean model: intensive computation of the tendency terms, a time-stepping routine, and for the parallel version, interleaved halo communication of the partition boundary. Indeed, this layout, and the lessons learned here, apply to almost all computational physics codes.

This work specifically tests unstructured horizontal meshes, as opposed to structured quadrilateral grids. Unstructured meshes refer to a neighbor's index using additional pointer arrays, so require an extra memory access for horizontal stencils. In structured grids, the physical neighbors are also neighbors in array space ($i+1, j+1$, etc), which leads to more contiguous memory access patterns that are easier for compilers to optimize. Our results show that unstructured meshes do not present any significant challenge in either Fortran or Julia. The use of a structured vertical index in the innermost position and testing with 100 layers provides sufficient contiguous memory access for cache locality.

In the end, we were impressed by our experience with Julia. It did fulfill the promise of fast and convenient prototyping, with the ability to eventually run at high speeds on multiple high performance architectures—after some effort and lessons learned by the developers. The Julia libraries for MPI and CUDA were powerful and convenient. E3SM does not have plans to develop model components with Julia, but this study provides a useful comparison to our Fortran and C++/C and codes as we move towards heterogeneous, exascale computers.

*Code and data availability.* Three code repositories were used for the performance comparisons in this study. These are publicly available on both GitHub and Zenodo:

1. Julia Shallow Water code for serial CPU, CUDA-GPU, and MPI-parallelized CPU

   GitHub: https://github.com/robertstrauss/MPAS_Ocean_Julia (license: GNU General Public License v3.0)

   Zenodo: https://doi.org/10.5281/zenodo.7493064 (license: Creative Commons Attribution 4.0 International)

2. Python Rotating Shallow Water Verification Suite

   GitHub: https://github.com/siddharthabishnu/Rotating_Shallow_Water_Verification_Suite.git. (license: LANL/UCAR*)
   This study used the specific code version https://github.com/siddharthabishnu/Rotating_Shallow_Water_Verification_Suite/tree/v1.0.1 (license: LANL/UCAR, https://github.com/MPAS-Dev/MPAS-Model/blob/master/LICENSE.)

   Zenodo: https://doi.org/10.5281/zenodo.7421135 (license: BSD 3-Clause "New" or "Revised")

3. Fortran-MPI MPAS Shallow Water code with Coastal Kelvin wave initial condition (Petersen et al., 2022)

   GitHub: https://github.com/MPAS-Dev/MPAS-Model. (license: LANL/UCAR, https://github.com/MPAS-Dev/MPAS-Model/blob/master/LICENSE.) This study used the specific code version https://github.com/mark-petersen/MPAS-Model/releases/tag/SW_julia_comparison_V1.0.

   Zenodo: https://doi.org/10.5281/zenodo.7439133 (license: Creative Commons Attribution 4.0 International)

The planar hexagonal MPAS-Ocean meshes used in this study for the numerical simulations and convergence tests of the coastal Kelvin wave and the inertia-gravity wave can be obtained from the Zenodo release of the Python Rotating Shallow Water Verification Suite Meshes at https://doi.org/10.5281/zenodo.7421135.

In order to reproduce the figures in this paper, follow the instructions below:

- Download the code for this project from (1) above. Acquire the necessary mesh files from https://doi.org/10.5281/zenodo.7421135, extract the zip file, and copy the:

  - 'MPAS_Ocean_Shallow_Water_Meshes/MPAS_Ocean_Shallow_Water_Meshes_Julia_Paper/InertiaGravityWaveMesh/' directory into the MPAS_Ocean_Julia repository at path 'MPAS_Ocean_Julia/';

  - 'MPAS_Ocean_Shallow_Water_Meshes/MPAS_Ocean_Shallow_Water_Meshes_Julia_Paper/CoastalKelvinWaveMesh/ConvergenceStudyMeshes/' directory into the MPAS_Ocean_Julia repository at path 'MPAS_Ocean_Julia/MPAS_O_Shallow_Water/'.

- Reproduce the figures in this paper as follows:

  - Figure 1: Run the Jupyter notebooks '/Operator_testing.ipynb' to generate the data for the convergence tests of the spatial operators, and '/operator_convergence_plotting.ipynb' to create plots from this data at '/output/operator_convergence/<operator>/Periodic/<figure>.pdf'. Run the notebook './InertiaGravityWaveConvergenceTest.ipynb' to generate the numerical solution and convergence plot of the inertia-gravity wave test case at './output/simulation_convergence/inertiagravitywave/Periodic/CPU/'.

  - Figures 2, 3, 4, and 5: On a cluster with at least 128 nodes and 64 processes per node, use the script './run_scaling_16x_to_512x.sh' to run the performance scaling tests on each mesh resolution starting from 16x16 all the way up to 512x512. The results will be saved in './output/kelvinwave/resolution<mesh size>/procs<maximum number of processors>/steps10/nvlevels100/'. Run the notebook '/GPU_performance.ipynb' on a node with an NVIDIA graphics card to initiate the performance tests on the GPU. Run

the notebook './scalingplots.ipynb' or the Julia script './scalingplots.jl' to generate the plots in the paper at '/plots/<type>/<figure>.pdf'.

- Tables 2 and 3: Run './serial_julia_performance.jl' with Julia to generate the timing data of the optimized Julia-CPU code. Download the unoptimized version of the code from https://github.com/robertstrauss/MPAS_Ocean_Julia/tree/unoptimized or MPAS_Ocean_Julia-unopt.zip from https://doi.org/10.5281/zenodo.7493064. Run the Julia script './serial_julia_performance.jl' in the directory of the unoptimized code. The results will be saved in text files at './output/serialCPU_timing/coastal_kelvinwave/unoptimized/steps_10/resolution_<mesh size>/' in the unoptimized directory and './output/serialCPU_timing/coastal_kelvinwave/steps_10/resolution_<mesh size>/' in the main/optimized directory.

*Author contributions.* Code development, testing, and timing were conducted by all authors. SB led the test case design and verification. RRS led the data analysis and Julia optimization. The manuscript was written cooperatively by all authors. MRP conceptualized the project and conducted funding acquisition.

*Competing interests.* The authors declare no competing interests.

*Acknowledgements.* SB was supported by Scientific Discovery through Advanced Computing (SciDAC) projects LEAP (Launching an Exascale ACME Prototype) and CANGA (Coupling Approaches for Next Generation Architectures) under the DOE Office of Science, Office of Biological and Environmental Research (BER). RRS gratefully acknowledges the support of the U.S. Department of Energy (DOE) through the Los Alamos National Laboratory (LANL) LDRD Program and the Center for Nonlinear Studies for this work. MRP was supported by the Energy Exascale Earth System Model (E3SM) project, also funded by the DOE BER.

This research used computational resources provided by: the Darwin testbed at LANL, which is funded by the Computational Systems and Software Environments subprogram of LANL's Advanced Simulation and Computing program (NNSA/DOE); the LANL Institutional Computing Program, which is supported by the DOE National Nuclear Security Administration under Contract No. 89233218CNA000001; and the National Energy Research Scientific Computing Center, a DOE Office of Science User Facility supported by the Office of Science of the DOE under Contract No. DE-AC02-05CH11231.

The authors extend their gratitude to the anonymous reviewers, whose valuable insights and constructive feedback were instrumental in elevating the quality and clarity of this paper. Additionally, the authors recognize the beneficial interactions with the CliMA team at Caltech and MIT, especially Simon Byrne, Milan Klöwer, Valentin Churavy, Gregory Wagner, Christopher Hill, Simone Silvestri, Navid Constantinou, and Andre Souza, alongside the E3SM team spread across multiple national laboratories within the United States. Their significant input has enriched the manner in which the paper has been articulated.

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
