# Peer review of "Comparing the Performance of Julia on CPUs versus GPUs and Julia-MPI versus Fortran-MPI: a case study with MPAS-Ocean (Version 7.1)"

_EGUsphere, 2023_

## Referee Comment (RC1)

**Review of "Comparing the Performance of Julia on CPUs versus GPUs and Julia-MPI versus Fortran-MPI: a case study with MPAS-Ocean (Version 7.1)" by Robert R. Strauss, Siddhartha Bishnu, and Mark R. Petersen**

**General comments**

The manuscript presents a shallow water model, written in the Julia programming language for CPUs and GPUs, and compares its performance to an object-oriented Python code and an established Fortran code. Since Julia is now emerging in the field of geophysical model development, and its application to unstructured-mesh PDE solvers is novel, this article is suitable for GMD. The manuscript is well written, and the model validation part is done very well. Unfortunately, the performance comparison study, which is one of the main parts of the manuscript, has major flaws. I can only recommend publication after substantial revisions are made to this part of the manuscript. Moreover, while I applaud the authors for providing code that is supposed to reproduce their results, I encountered several issues when trying to run it.

This reviewer is familiar with Julia, but many potential readers won't be. The Julia compilation model, which is the key to its performance, is first briefly described in the results section and further discussed in the optimization tips section. These descriptions are not fully correct (see the specific comments below), and could be made more clear. I think it would be a good idea to centralize this material and provide a short introduction to Julia in the methods section, emphasizing how its compilation model enables high-performance computing and how static code generation facilitates GPU computations.

The authors obtained very impressive speed-ups from using GPUs - up to 100,000x for the computation time. Their result that the GPU computation times for their model do not depend on problem size is very surprising and, frankly, suspicious. Looking at the provided code, the authors seem to profile their model using the Julia macros `@elapsed` and `@benchmark`. However, based on the CUDA.jl profiling guide[1], this is not the right way to profile GPU code, which launches kernels asynchronously. The correct way is to use `CUDA.@elapsed` and add synchronization to functions profiled by `@benchmark`. Otherwise, only the cost of launching kernels will be timed and not the actual cost of computations. If this is how the GPU profiling was done, then the GPU benchmarks must be repeated and the manuscript revised based on the new results.

In general, while many speed-up numbers are provided in the paper, there is no rigorous discussion of the obtained values. These values should be put in the context of known hardware capabilities. MPAS-Ocean is a well-established code, and its performance bottlenecks are surely known. Since it is a low-order finite-volume code, most likely the main performance limiter is memory bandwidth. The authors provided many details about the hardware they used, such as peak flops, the number of CUDA cores, and cache sizes. However, the memory bandwidth value is only provided for the RTX 8000 GPU, but not the CPU. It is also unclear whether it was measured empirically, or taken from the vendor specification. Please focus on the hardware characteristics that are relevant to the model computational performance and frame the discussion of speed-ups around these characteristics.

While there are many ways to compare performance of heterogeneous systems, presenting comparisons of a single CPU core to a single GPU is not a valid practice. As the aim of the paper is to demonstrate the suitability of Julia for HPC, comparisons between a full CPU node and a GPU should be done, while perhaps noting their TDP values. As an MPI version of the code was created, this should be straightforward.

There is a number of possibilities to extend the performance analysis part of the paper, which, while not absolutely necessary, would significantly strengthen it. Since the number of kernels executed by each model time step is small, this present an opportunity to use profilers to do performance analysis at the level of individual kernels. Roofline plots could be created. The scaling study could also be extended to cover weak scaling.
* * *
[1] https://cuda.juliagpu.org/stable/development/profiling/

**Specific comments**

- Lines 38-40: "In recent years, shallow water solvers such as Oceananigans.jl (Ramadhan et al., 2020) and ShallowWaters.jl (Klöwer et al., 2022) have been developed in Julia. These codes (…) are equipped with capabilities for running on GPUs to achieve high performance." According to its documentation `Oceananigans.jl` is not just a shallow water solver, but can solve nonhydrostatic and hydrostatic Boussinesq equations. There is no mention that `ShallowWaters.jl` can run on GPUs in its documentation, and its `Project.toml` doesn't include any GPU packages.
- Line 77, equation (2a): shouldn't the kinetic energy term be under the gradient operator ?
- Line 82: (2b) is referred to as the discrete momentum equation. Shouldn't this be (2a) ?
- Lines 278-279: "making all types and subtypes concrete rather than abstract, to minimize on-the-fly compilation" How does making the types concrete minimize compilation ? Julia performs just-in-time compilation regardless of whether the supplied arguments are concrete or abstract. The cost of abstract types is the cost of missed optimizations and additional runtime dispatch.
- Lines 288-289: "In addition, single-precision floating point numbers (CUDA Float32 data type) calculations may execute significantly faster than Float64 (Julia Development Team, a)." It is true that most customer-grade GPUs have limited double-precision capabilities. This is usually not the case for GPUs targeting the HPC market. However, if the presented code is bandwidth limited, wouldn't the maximum possible speed-up from switching to Float32 be 2x ?
- Figure 3.: The title mentions occupancy, but only execution times are shown. Which kernel was profiled ? Are the results the same for other kernels ? It would be better if the x axis ticks showed some typical block size values, such as 32, 64, 128, 256, 512, and 1024.
- Lines 330-333 and Figure 4.: Any idea why Fortran-MPI computations scale worse than Julia ?
- Lines 336-347: Again, I found this discussion of the Julia compilation model not fully correct. Julia does not require variables with full type declaration to achieve fast code, since it performs aggressive type inference. The authors discuss an issue that only concerns type declarations of struct members. Moreover, the authors again seem to suggest that just-in-time compilation occurs only for `Any` or abstract types, which is not correct. Please revise this paragraph. As mentioned in the general remarks, some of the material regarding the compilation model should probably be presented earlier in the text.

**Technical comments**

- Some words are not capitalized consistently and correctly (ssh and SSH, python and Python, Numpy and NumPy).
- Line 9 in the abstract: "The GPU-accelerated Julia code is attained a speed-up" – spurious "is".
- Line 153: `pressueGradient` should be `sshGradient`.
- I was not able to run the provided code as is. I only tried to run the optimized version. The optimized code contains typos and references to undefined variables. Some examples of the problems I encountered:
    - in `GPU_CPU_performance_comparison_meshes.ipynb` the file `cuda_time_steppers.jl` is not included, which makes CUDA tendency functions undefined.
    - In the same file `calculate_ssh_tendency_cuda!` is used, but it is not defined in `cuda_tendencies.jl`, or anywhere else.
    - In `calculate_normal_velocity_tendency_cuda!` there is a typo in `mpasOean.maxLevelEdgeTop`. Moreover, `maxLevelEdgeTop` is not a member of the `MPAS_Ocean_CUDA` struct. There are more issues along the same lines. Please provide a version of the optimized code that can be run.
- I had some issues with obtaining the mesh files used in this study. The readme file points to a Zenodo archive[2]. However, the article uses meshes of size 128x128, 256x256, and 512x512. The Zenodo archive contains meshes of size 64x64, 96x96, 144x144, 216x216, and 324x324. Where can I find the mesh files used in this study ?
* * *
[2] https://zenodo.org/record/7419817#.Y63p4C-B1pQ

---

## Author Comment (AC1)

**Reviewer 1: Revision of Manuscript, "Julia for Geophysical Fluid Dynamics: Performance Comparisons between CPU, GPU, and Fortran-MPI"**

The authors are grateful for the reviewers' insightful suggestions, which have contributed to the improvement of the manuscript. The major change is that the Julia-GPU, Julia-MPI, Fortran-MPI simulations have all been retested on Perlmutter at NERSC, so that comparisons can be made directly between single-node CPU and single-node GPU. Perlmutter uses AMD EPYC 7763 CPU nodes and NVIDIA A100 GPUs, so is newer and more relevant to readers than the previous tests on Cori. We have also added more introductory material on Julia, a longer results and discussion section, weak scaling plots, and plots that directly compare full-node CPU and GPU performance.

**Major Comments:**

1. The manuscript presents a shallow water model, written in the Julia programming language for CPUs and GPUs, and compares its performance to an object-oriented Python code and an established Fortran code. Since Julia is now emerging in the field of geophysical model development, and its application to unstructured-mesh PDE solvers is novel, this article is suitable for GMD. The manuscript is well written, and the model validation part is done very well. Unfortunately, the performance comparison study, which is one of the main parts of the manuscript, has major flaws. I can only recommend publication after substantial revisions are made to this part of the manuscript. Moreover, while I applaud the authors for providing code that is supposed to reproduce their results, I encountered several issues when trying to run it.

   **Response:** We have revised the manuscript based on the suggestions provided, and fixed the issues with the accompanying code to ensure reproducibility.

2. This reviewer is familiar with Julia, but many potential readers won't be. The Julia compilation model, which is the key to its performance, is first briefly described in the results section and further discussed in the optimization tips section. These descriptions are not fully correct (see the specific comments below), and could be made more clear. I think it would be a good idea to centralize this material and provide a short introduction to Julia in the methods section, emphasizing how its compilation model enables high-performance computing and how static code generation facilitates GPU computations.

   **Response:** This is a good suggestion. We have added a new section 2, which is an introduction to Julia. We have also corrected the text from your specific comments below. Thank you.

3. The authors obtained very impressive speed-ups from using GPUs - up to 100,000x for the computation time. Their result that the GPU computation times for their model do not depend on problem size is very surprising and, frankly, suspicious. Looking at the provided code, the authors seem to profile their model using the Julia macros @elapsed and @benchmark. However, based on the CUDA.jl profiling guide1, this is not the right way to profile GPU code, which launches kernels asynchronously. The correct way is to use CUDA.@elapsed and add synchronization to functions profiled by @benchmark. Otherwise, only the cost of launching kernels will be timed and not the actual cost of computations. If this is how the GPU profiling was done, then the GPU benchmarks must be repeated and the manuscript revised based on the new results.

   **Response:** Thank you for this incredibly valuable suggestion. After implementing the appropriate CUDA macros to accurately measure execution time on the GPUs, we discovered that the previous speed-up times were indeed incorrect. The rectified results, depicted in Figure 2, present a more reasonable scenario, where the scaling of the GPU closely aligns with that of a full node comprising 64 processes.

4. In general, while many speed-up numbers are provided in the paper, there is no rigorous discussion of the obtained values. These values should be put in the context of known hardware capabilities. MPAS-Ocean is a well-established code, and its performance bottlenecks are surely known. Since it is a low-order finite-volume code, most likely the main performance limiter is memory bandwidth.

The authors provided many details about the hardware they used, such as peak flops, the number of CUDA cores, and cache sizes. However, the memory bandwidth value is only provided for the RTX 8000 GPU, but not the CPU. It is also unclear whether it was measured empirically, or taken from the vendor specification. Please focus on the hardware characteristics that are relevant to the model computational performance and frame the discussion of speed-ups around these characteristics.

**Response:** Please see the new section 3.7, Hardware and Compiler Specifications. Table 1 has been added to describe the hardware, which is now a CPU node versus a GPU node of Perlmutter at NERSC.

5. While there are many ways to compare performance of heterogeneous systems, presenting comparisons of a single CPU core to a single GPU is not a valid practice. As the aim of the paper is to demonstrate the suitability of Julia for HPC, comparisons between a full CPU node and a GPU should be done, while perhaps noting their TDP values. As an MPI version of the code was created, this should be straightforward.

**Response:** After rectifying our GPU timing measurements, we proceeded to evaluate the performance of our Fortran and Julia codes on the NVIDIA TESLA A100 GPU. We compared this performance to the execution on a full CPU node consisting of 64 processes. The wall-clock times for both scenarios, illustrated in Figure 2, demonstrate remarkably similar results.

6. There is a number of possibilities to extend the performance analysis part of the paper, which, while not absolutely necessary, would significantly strengthen it. Since the number of kernels executed by each model time step is small, this presents an opportunity to use profilers to do performance analysis at the level of individual kernels. Roofline plots could be created. The scaling study could also be extended to cover weak scaling.

**Response:** We have expanded the performance analysis section of the paper to encompass weak scaling. In order to identify lines of code that are consuming excessive amount of time on the CPU, we employed line-by-line profilers and allocation trackers (as mentioned in the section on optimization tips for Julia developers).

**Specific Comments**:

1. Lines 38-40: "In recent years, shallow water solvers such as Oceananigans.jl (Ramadhan et al., 2020) and ShallowWaters.jl (Klöwer et al., 2022) have been developed in Julia. These codes (...) are equipped with capabilities for running on GPUs to achieve high performance." According to its documentation, Oceananigans.jl is not just a shallow water solver, but can solve nonhydrostatic and hydrostatic Boussinesq equations. There is no mention that ShallowWaters.jl can run on GPUs in its documentation, and its Project.toml doesn't include any GPU packages.

**Response:** The text has been updated with this information.

2. Line 77, Equation (2a): Shouldn't the kinetic energy term be under the gradient operator?

**Response:** Yes, we have fixed it.

3. Line 82: (2b) is referred to as the discrete momentum equation. Shouldn't this be (2a)?

**Response:** Yes, we have fixed it.

4. Lines 278-279: "making all types and subtypes concrete rather than abstract, to minimize on-the-fly compilation". How does making the types concrete minimize compilation? Julia performs just-in-time compilation regardless of whether the supplied arguments are concrete or abstract. The cost of abstract types is the cost of missed optimizations and additional runtime dispatch.

**Response:** Thank you for highlighting the flaw in our explanation. We have duly corrected these lines and enhanced our clarification on these points.

5. Lines 288-289: "In addition, single-precision floating point numbers (CUDA Float32 data type) calculations may execute significantly faster than Float64 (Julia Development Team)." It is true that

most customer-grade GPUs have limited double-precision capabilities. This is usually not the case for GPUs targeting the HPC market. However, if the presented code is bandwidth limited, wouldn't the maximum possible speed-up from switching to Float32 be 2x?

**Response:** The sentence mentioned above has been removed from our revised manuscript. In the updated version, we evaluate the performance of the CUDA code on the NVIDIA TESLA A100 GPU specifically designed for high-performance computing (HPC). Moreover, we employ double precision throughout our analysis, rendering the previously obtained results with single precision irrelevant.

6. Figure 3: The title mentions occupancy, but only execution times are shown. Which kernel was profiled? Are the results the same for other kernels? It would be better if the x axis ticks showed some typical block size values, such as 32, 64, 128, 256, 512, and 1024.

**Response:** The mentioned figure has been excluded from our revised manuscript. Upon properly timing the GPU computations, we realised that this particular result is neither accurate nor pertinent to our analysis.

7. Lines 330-333 and Figure 4.: Any idea why Fortran-MPI computations scale worse than Julia?

**Response:** In the revised plots using Perlmutter comparing Fortran-MPI and Julia-MPI, Fortran performance is much closer to Julia, but still a bit slower. We discussed this with performance experts at NERSC, but did not find an explanation.

8. Lines 336-347: Again, I found this discussion of the Julia compilation model not fully correct. Julia does not require variables with full type declaration to achieve fast code, since it performs aggressive type inference. The authors discuss an issue that only concerns type declarations of struct members. Moreover, the authors again seem to suggest that just-in-time compilation occurs only for Any or abstract types, which is not correct. Please revise this paragraph. As mentioned in the general remarks, some of the material regarding the compilation model should probably be presented earlier in the text.

**Response:** Thanks to the clarity of your feedback, we have made appropriate corrections to our explanation on the Julia compilation model. We have explicitly specified that the performance boost associated with concrete type definitions only apply to structs. Moreover, we have relocated the comprehensive content on the Julia compilation model to Section 2, which serves as an introductory segment on Julia.

**Technical Comments**:

1. Some words are not capitalized consistently and correctly (ssh and SSH, python and Python, Numpy and NumPy).

   **Response:** Fixed.

2. Line 9 in the abstract: "The GPU-accelerated Julia code is attained a speed-up"

   **Response:** Fixed.

3. spurious "is".

   **Response:** Removed.

4. Line 153: pressureGradient should be sshGradient.

   **Response:** Fixed.

5. I was not able to run the provided code as is. I only tried to run the optimized version. The optimized code contains typos and references to undefined variables. Some examples of the problems I encountered:

   (a) in `GPU_CPU_performance_comparison_meshes.ipynb` the file `cuda_time_steppers.jl` is not included, which makes CUDA tendency functions undefined.

**Response:** Fixed.

(b) In the same file `calculate_ssh_tendency_cuda!` is used, but it is not defined in `cuda_tendencies.jl`, or anywhere else.

**Response:** Fixed.

(c) In `calculate_normal_velocity_tendency_cuda!` there is a typo in mpasOean.maxLevelEdgeTop.

**Response:** Fixed.

6. Moreover, maxLevelEdgeTop is not a member of the `MPAS_Ocean_CUDA` struct. There are more issues along the same lines. Please provide a version of the optimized code that can be run.

**Response:** In our initial submission, the MPAS_Ocean_CUDA struct was derived from an outdated version of the code, predating our implementation of the multi-layered shallow water equations, where the relevance of maxLevelEdgeTop comes into play. However, we have since updated the code, ensuring that the MPAS_Ocean_CUDA struct now contains all the essential fields required for computation.

7. I had some issues with obtaining the mesh files used in this study. The readme file points to a Zenodo archive. However, the article uses meshes of size 128x128, 256x256, and 512x512. The Zenodo archive contains meshes of size 64x64, 96x96, 144x144, 216x216, and 324x324. Where can I find the mesh files used in this study?

**Response:** In our revised manuscript, we showcase the convergence of the spatial operators on the same set of meshes used for obtaining convergence of the numerical solution of the inertia gravity wave test case. They can be obtained from either

(a) MPAS_Ocean_Shallow_Water_Meshes_Convergence_Study/Periodic, or

(b) InertiaGravityWaveMesh/ConvergenceStudyMeshes

on the Zenodo archive.

---

## Author Comment (AC2)

**Reviewer 2: Revision of Manuscript, "Julia for Geophysical Fluid Dynamics: Performance Comparisons between CPU, GPU, and Fortran-MPI"**

The authors are grateful for the reviewers' insightful suggestions, which have contributed to the improvement of the manuscript. The major change is that the Julia-GPU, Julia-MPI, Fortran-MPI simulations have all been retested on Perlmutter at NERSC, so that comparisons can be made directly between single-node CPU and single-node GPU. Perlmutter uses AMD EPYC 7763 CPU nodes and NVIDIA A100 GPUs, so is newer and more relevant to readers than the previous tests on Cori. We have also added more introductory material on Julia, a longer results and discussion section, weak scaling plots, and plots that directly compare full-node CPU and GPU performance.

**Major Comments:** The authors have made significant contributions by developing a shallow water solver using Julia language and comparing its performance with a solver written in Fortran. Furthermore, they have successfully implemented their solver on a GPU, demonstrating a remarkable speed-up. While the overall results appear promising, I would suggest considering the following points to further enhance the paper:

1. In section 3.2, it would greatly enhance the paper to include a table comparing the specifications of the CPU and GPU used in the simulations. This table should provide a comprehensive comparison of various factors, such as FLOPS (Floating-Point Operations Per Second) and memory bandwidth, specifically for both 32-bit and 64-bit computations. Additionally, it would be valuable to summarize the versions of the toolchain that were utilized during these computations. This information will provide readers with a better understanding of the hardware and software environment in which the simulations were conducted, allowing for a more comprehensive evaluation of the results.

   **Response:** Please see the new section 3.7, Hardware and Compiler Specifications. Table 1 has been added to describe the hardware, which is now a compute node versus a GPU node of Perlmutter at NERSC. Several paragraphs were added on the toolchain for both Fortran and Julia.

2. In section 3.2, it would be beneficial to include a comparison of the performance between the Julia code and the Fortran code in a single-core execution. This comparison will provide readers with insights into the optimization of the Julia code for serial computation.

   **Response:** We have extended the comparison between Fortran and Julia to a single core, with the results illustrated in the strong scaling plots in Figure 3.

3. In Section 3.2, the authors mentioned that all codes were executed in double precision and highlighted the faster simulation on the NVIDIA RTX8000 GPU compared to the CPU. However, it is important to consider that the RTX8000 is primarily designed for consumer applications and may exhibit slower performance in double precision computation. To provide a more comprehensive evaluation, it would be valuable to compare the computation on a high-performance computing (HPC) targeted GPU, such as the NVIDIA TESLA A100, which is known for their robust performance in double precision computation and are specifically designed to excel in HPC workloads. Otherwise, please compare all simulations in single precision.

   **Response:** This is a great suggestion. We switched all of our performance comparisons to Perlmutter at NERSC, which came on line this year. We chose Perlmutter in order to use an HPC-targeted GPU, the NVIDIA TESLA A100, as suggested here.

4. In section 3.3, it is evident that Julia-MPI outperformed Fortran-MPI in terms of computation, but it took more time for communication. To provide a clearer understanding of the experimental setup, it would be beneficial to specify the Fortran compiler and Julia interpreters, along with the related toolchain, that were employed in the study. Additionally, it is important to mention the specific version of the MPI library used for both the Fortran-MPI and Julia-MPI implementations. This information will help readers better comprehend the underlying MPI libraries utilized in each case and the potential impact they may have had on the communication performance.

**Response:** Please see the paragraphs on toolchains for Fortran and Julia, added to the new section 3.7.

Moreover, it is worth exploring the possibility that different MPI libraries might have been employed for the Fortran and Julia codes. If this is the case, it should be explicitly stated in the paper, along with the versions of the MPI libraries used for each implementation. Clarifying this aspect will enable readers to consider any discrepancies or optimizations associated with the MPI libraries employed in the Fortran and Julia implementations.

**Response:** We have added the MPICH version to the paper in section 4.3, which is version 4.0 for Julia and 3.4 for Fortran. Unfortunately, we were not able to compare with other versions as we were limited to the modules available on Perlmutter.

5. I think hyper threading may be disabled in supercomputer. It would be helpful to omit the hyper-thread performance of the CPU in section 3.3.

   **Response:** We have removed the section on hyper-threading performance.

---

## Referee Report (RR1)

**Review of "Julia for Geophysical Fluid Dynamics: Performance Comparisons between CPU, GPU, and Fortran-MPI" by Robert R. Strauss, Siddhartha Bishnu, and Mark R. Petersen**

**General comments**

This is a revised submission. The authors have addressed most of my comments. I especially welcome the new "Julia in a Nutshell" section, which is overall very well done. I recommend acceptance after minor revisions.

**Minor comments**

- Line 74: Julia's "superior memory management": I don't understand what the authors mean here. Julia is a garbage collected language which trades off programmer control for ease of memory management. Garbage collection can have undesirable effects in parallel applications (see for example this Julia issue https://github.com/JuliaLang/julia/issues/49316). It is not strictly superior to other memory management mechanisms.
- Line 80: "Julia has found widespread application ...": I think this statement is too strong. Julia definitely does not have "widespread" application in web development, and my impression is that it is only gaining grounds in the other fields mentioned.
- Line 126: Strictly speaking `Array` is not a concrete type since it is parametric (this can be easily checked in Julia by evaluating `isconcretetype(Array)` which returns `false`). Additionally, here and in other places Julia types are written in plain text, but other times they are displayed as code. Please unify the style throughout the manuscript.
- There is a typo in equation (1a) (there should be only one gradient operator).
- I have a couple of suggestions for Table 1. Please mention that the TDP values are per CPU and per GPU. The A100 GPU comes in two variants with TDP of 300 W or 400 W. I believe that Perlmutter has the 400 W version. To be consistent with the GPU specification, please say that the CPU flops are for double precision.
- Lines 510-514: "Based on technical specification ...": The comparison of flops values implies that the authors think that their code is compute bound. Based on the low number of operations I would expect the code to be bandwidth limited and the bandwidth ratio to be a more appropriate speedup bound, at least for large problem sizes. Can the authors comment on this ?
- Line 618 "Julia-GPU scaled very similarly to Julia-MPI": Maybe "performed" would be better here than "scaled" ?
- Line 622: "(...) and sample results rarely, GPUs can offer significant speed-ups": This might be true in theory, but I don't see how the presented results support that conclusion. Looking at Figure 2, even in the "Computation Only" plot the CPU is faster than the GPU, even though the GPU is theoretically much more powerful.

---

## Author Response (AR3)

**Reviewer 1: First Revision of Manuscript, "Julia for Geophysical Fluid Dynamics: Performance Comparisons between CPU, GPU, and Fortran-MPI"**

The authors are grateful for the reviewers' insightful suggestions, which have contributed to the improvement of the manuscript. The major change is that the Julia-GPU, Julia-MPI, Fortran-MPI simulations have all been retested on Perlmutter at NERSC, so that comparisons can be made directly between single-node CPU and single-node GPU. Perlmutter uses AMD EPYC 7763 CPU nodes and NVIDIA A100 GPUs, so is newer and more relevant to readers than the previous tests on Cori. We have also added more introductory material on Julia, a longer results and discussion section, weak scaling plots, and plots that directly compare full-node CPU and GPU performance.

**Major Comments:**

1. The manuscript presents a shallow water model, written in the Julia programming language for CPUs and GPUs, and compares its performance to an object-oriented Python code and an established Fortran code. Since Julia is now emerging in the field of geophysical model development, and its application to unstructured-mesh PDE solvers is novel, this article is suitable for GMD. The manuscript is well written, and the model validation part is done very well. Unfortunately, the performance comparison study, which is one of the main parts of the manuscript, has major flaws. I can only recommend publication after substantial revisions are made to this part of the manuscript. Moreover, while I applaud the authors for providing code that is supposed to reproduce their results, I encountered several issues when trying to run it.

    **Response:** We have revised the manuscript based on the suggestions provided, and fixed the issues with the accompanying code to ensure reproducibility.

2. This reviewer is familiar with Julia, but many potential readers won't be. The Julia compilation model, which is the key to its performance, is first briefly described in the results section and further discussed in the optimization tips section. These descriptions are not fully correct (see the specific comments below), and could be made more clear. I think it would be a good idea to centralize this material and provide a short introduction to Julia in the methods section, emphasizing how its compilation model enables high-performance computing and how static code generation facilitates GPU computations.

    **Response:** This is a good suggestion. We have added a new section 2, which is an introduction to Julia. We have also corrected the text from your specific comments below. Thank you.

3. The authors obtained very impressive speed-ups from using GPUs - up to 100,000x for the computation time. Their result that the GPU computation times for their model do not depend on problem size is very surprising and, frankly, suspicious. Looking at the provided code, the authors seem to profile their model using the Julia macros @elapsed and @benchmark. However, based on the CUDA.jl profiling guide1, this is not the right way to profile GPU code, which launches kernels asynchronously. The correct way is to use CUDA.@elapsed and add synchronization to functions profiled by @benchmark. Otherwise, only the cost of launching kernels will be timed and not the actual cost of computations. If this is how the GPU profiling was done, then the GPU benchmarks must be repeated and the manuscript revised based on the new results.

    **Response:** Thank you for this incredibly valuable suggestion. After implementing the appropriate CUDA macros to accurately measure execution time on the GPUs, we discovered that the previous speed-up times were indeed incorrect. The rectified results, depicted in Figure 2, present a more reasonable scenario, where the scaling of the GPU closely aligns with that of a full node comprising 64 processes.

4. In general, while many speed-up numbers are provided in the paper, there is no rigorous discussion of the obtained values. These values should be put in the context of known hardware capabilities. MPAS-Ocean is a well-established code, and its performance bottlenecks are surely known. Since it is a low-order finite-volume code, most likely the main performance limiter is memory bandwidth.

The authors provided many details about the hardware they used, such as peak flops, the number of CUDA cores, and cache sizes. However, the memory bandwidth value is only provided for the RTX 8000 GPU, but not the CPU. It is also unclear whether it was measured empirically, or taken from the vendor specification. Please focus on the hardware characteristics that are relevant to the model computational performance and frame the discussion of speed-ups around these characteristics.

**Response:** Please see the new section 3.7, Hardware and Compiler Specifications. Table 1 has been added to describe the hardware, which is now a CPU node versus a GPU node of Perlmutter at NERSC.

5. While there are many ways to compare performance of heterogeneous systems, presenting comparisons of a single CPU core to a single GPU is not a valid practice. As the aim of the paper is to demonstrate the suitability of Julia for HPC, comparisons between a full CPU node and a GPU should be done, while perhaps noting their TDP values. As an MPI version of the code was created, this should be straightforward.

**Response:** After rectifying our GPU timing measurements, we proceeded to evaluate the performance of our Fortran and Julia codes on the NVIDIA TESLA A100 GPU. We compared this performance to the execution on a full CPU node consisting of 64 processes. The wall-clock times for both scenarios, illustrated in Figure 2, demonstrate remarkably similar results.

6. There is a number of possibilities to extend the performance analysis part of the paper, which, while not absolutely necessary, would significantly strengthen it. Since the number of kernels executed by each model time step is small, this presents an opportunity to use profilers to do performance analysis at the level of individual kernels. Roofline plots could be created. The scaling study could also be extended to cover weak scaling.

**Response:** We have expanded the performance analysis section of the paper to encompass weak scaling. In order to identify lines of code that are consuming excessive amount of time on the CPU, we employed line-by-line profilers and allocation trackers (as mentioned in the section on optimization tips for Julia developers).

**Specific Comments**:

1. Lines 38-40: "In recent years, shallow water solvers such as Oceananigans.jl (Ramadhan et al., 2020) and ShallowWaters.jl (Klöwer et al., 2022) have been developed in Julia. These codes (. . .) are equipped with capabilities for running on GPUs to achieve high performance." According to its documentation, Oceananigans.jl is not just a shallow water solver, but can solve nonhydrostatic and hydrostatic Boussinesq equations. There is no mention that ShallowWaters.jl can run on GPUs in its documentation, and its Project.toml doesn't include any GPU packages.

**Response:** The text has been updated with this information.

2. Line 77, Equation (2a): Shouldn't the kinetic energy term be under the gradient operator?

**Response:** Yes, we have fixed it.

3. Line 82: (2b) is referred to as the discrete momentum equation. Shouldn't this be (2a)?

**Response:** Yes, we have fixed it.

4. Lines 278-279: "making all types and subtypes concrete rather than abstract, to minimize on-the-fly compilation". How does making the types concrete minimize compilation? Julia performs just-in-time compilation regardless of whether the supplied arguments are concrete or abstract. The cost of abstract types is the cost of missed optimizations and additional runtime dispatch.

**Response:** Thank you for highlighting the flaw in our explanation. We have duly corrected these lines and enhanced our clarification on these points.

5. Lines 288-289: "In addition, single-precision floating point numbers (CUDA Float32 data type) calculations may execute significantly faster than Float64 (Julia Development Team)." It is true that

most customer-grade GPUs have limited double-precision capabilities. This is usually not the case for GPUs targeting the HPC market. However, if the presented code is bandwidth limited, wouldn't the maximum possible speed-up from switching to Float32 be 2x?

**Response:** The sentence mentioned above has been removed from our revised manuscript. In the updated version, we evaluate the performance of the CUDA code on the NVIDIA TESLA A100 GPU specifically designed for high-performance computing (HPC). Moreover, we employ double precision throughout our analysis, rendering the previously obtained results with single precision irrelevant.

6. Figure 3: The title mentions occupancy, but only execution times are shown. Which kernel was profiled? Are the results the same for other kernels? It would be better if the x axis ticks showed some typical block size values, such as 32, 64, 128, 256, 512, and 1024.

**Response:** The mentioned figure has been excluded from our revised manuscript. Upon properly timing the GPU computations, we realised that this particular result is neither accurate nor pertinent to our analysis.

7. Lines 330-333 and Figure 4.: Any idea why Fortran-MPI computations scale worse than Julia?

**Response:** In the revised plots using Perlmutter comparing Fortran-MPI and Julia-MPI, Fortran performance is much closer to Julia, but still a bit slower. We discussed this with performance experts at NERSC, but did not find an explanation.

8. Lines 336-347: Again, I found this discussion of the Julia compilation model not fully correct. Julia does not require variables with full type declaration to achieve fast code, since it performs aggressive type inference. The authors discuss an issue that only concerns type declarations of struct members. Moreover, the authors again seem to suggest that just-in-time compilation occurs only for Any or abstract types, which is not correct. Please revise this paragraph. As mentioned in the general remarks, some of the material regarding the compilation model should probably be presented earlier in the text.

**Response:** Thanks to the clarity of your feedback, we have made appropriate corrections to our explanation on the Julia compilation model. We have explicitly specified that the performance boost associated with concrete type definitions only apply to structs. Moreover, we have relocated the comprehensive content on the Julia compilation model to Section 2, which serves as an introductory segment on Julia.

**Technical Comments**:

1. Some words are not capitalized consistently and correctly (ssh and SSH, python and Python, Numpy and NumPy).

   **Response:** Fixed.

2. Line 9 in the abstract: "The GPU-accelerated Julia code is attained a speed-up"

   **Response:** Fixed.

3. spurious "is".

   **Response:** Removed.

4. Line 153: pressureGradient should be sshGradient.

   **Response:** Fixed.

5. I was not able to run the provided code as is. I only tried to run the optimized version. The optimized code contains typos and references to undefined variables. Some examples of the problems I encountered:

   (a) in `GPU_CPU_performance_comparison_meshes.ipynb` the file `cuda_time_steppers.jl` is not included, which makes CUDA tendency functions undefined.

**Response:** Fixed.

(b) In the same file `calculate_ssh_tendency_cuda!` is used, but it is not defined in `cuda_tendencies.jl`, or anywhere else.

**Response:** Fixed.

(c) In `calculate_normal_velocity_tendency_cuda!` there is a typo in mpasOean.maxLevelEdgeTop.

**Response:** Fixed.

6. Moreover, maxLevelEdgeTop is not a member of the `MPAS_Ocean_CUDA` struct. There are more issues along the same lines. Please provide a version of the optimized code that can be run.

**Response:** In our initial submission, the MPAS_Ocean_CUDA struct was derived from an outdated version of the code, predating our implementation of the multi-layered shallow water equations, where the relevance of maxLevelEdgeTop comes into play. However, we have since updated the code, ensuring that the MPAS_Ocean_CUDA struct now contains all the essential fields required for computation.

7. I had some issues with obtaining the mesh files used in this study. The readme file points to a Zenodo archive. However, the article uses meshes of size 128x128, 256x256, and 512x512. The Zenodo archive contains meshes of size 64x64, 96x96, 144x144, 216x216, and 324x324. Where can I find the mesh files used in this study?

**Response:** In our revised manuscript, we showcase the convergence of the spatial operators on the same set of meshes used for obtaining convergence of the numerical solution of the inertia gravity wave test case. They can be obtained from either

(a) MPAS_Ocean_Shallow_Water_Meshes_Convergence_Study/Periodic, or

(b) InertiaGravityWaveMesh/ConvergenceStudyMeshes

on the Zenodo archive.

**Reviewer 2: First Revision of Manuscript, "Julia for Geophysical Fluid Dynamics: Performance Comparisons between CPU, GPU, and Fortran-MPI"**

The authors are grateful for the reviewers' insightful suggestions, which have contributed to the improvement of the manuscript. The major change is that the Julia-GPU, Julia-MPI, Fortran-MPI simulations have all been retested on Perlmutter at NERSC, so that comparisons can be made directly between single-node CPU and single-node GPU. Perlmutter uses AMD EPYC 7763 CPU nodes and NVIDIA A100 GPUs, so is newer and more relevant to readers than the previous tests on Cori. We have also added more introductory material on Julia, a longer results and discussion section, weak scaling plots, and plots that directly compare full-node CPU and GPU performance.

**Major Comments:** The authors have made significant contributions by developing a shallow water solver using Julia language and comparing its performance with a solver written in Fortran. Furthermore, they have successfully implemented their solver on a GPU, demonstrating a remarkable speed-up. While the overall results appear promising, I would suggest considering the following points to further enhance the paper:

1. In section 3.2, it would greatly enhance the paper to include a table comparing the specifications of the CPU and GPU used in the simulations. This table should provide a comprehensive comparison of various factors, such as FLOPS (Floating-Point Operations Per Second) and memory bandwidth, specifically for both 32-bit and 64-bit computations. Additionally, it would be valuable to summarize the versions of the toolchain that were utilized during these computations. This information will provide readers with a better understanding of the hardware and software environment in which the simulations were conducted, allowing for a more comprehensive evaluation of the results.

   **Response:** Please see the new section 3.7, Hardware and Compiler Specifications. Table 1 has been added to describe the hardware, which is now a compute node versus a GPU node of Perlmutter at NERSC. Several paragraphs were added on the toolchain for both Fortran and Julia.

2. In section 3.2, it would be beneficial to include a comparison of the performance between the Julia code and the Fortran code in a single-core execution. This comparison will provide readers with insights into the optimization of the Julia code for serial computation.

   **Response:** We have extended the comparison between Fortran and Julia to a single core, with the results illustrated in the strong scaling plots in Figure 3.

3. In Section 3.2, the authors mentioned that all codes were executed in double precision and highlighted the faster simulation on the NVIDIA RTX8000 GPU compared to the CPU. However, it is important to consider that the RTX8000 is primarily designed for consumer applications and may exhibit slower performance in double precision computation. To provide a more comprehensive evaluation, it would be valuable to compare the computation on a high-performance computing (HPC) targeted GPU, such as the NVIDIA TESLA A100, which is known for their robust performance in double precision computation and are specifically designed to excel in HPC workloads. Otherwise, please compare all simulations in single precision.

   **Response:** This is a great suggestion. We switched all of our performance comparisons to Perlmutter at NERSC, which came on line this year. We chose Perlmutter in order to use an HPC-targeted GPU, the NVIDIA TESLA A100, as suggested here.

4. In section 3.3, it is evident that Julia-MPI outperformed Fortran-MPI in terms of computation, but it took more time for communication. To provide a clearer understanding of the experimental setup, it would be beneficial to specify the Fortran compiler and Julia interpreters, along with the related toolchain, that were employed in the study. Additionally, it is important to mention the specific version of the MPI library used for both the Fortran-MPI and Julia-MPI implementations. This information will help readers better comprehend the underlying MPI libraries utilized in each case and the potential impact they may have had on the communication performance.

**Response:** Please see the paragraphs on toolchains for Fortran and Julia, added to the new section 3.7.

Moreover, it is worth exploring the possibility that different MPI libraries might have been employed for the Fortran and Julia codes. If this is the case, it should be explicitly stated in the paper, along with the versions of the MPI libraries used for each implementation. Clarifying this aspect will enable readers to consider any discrepancies or optimizations associated with the MPI libraries employed in the Fortran and Julia implementations.

**Response:** We have added the MPICH version to the paper in section 4.3, which is version 4.0 for Julia and 3.4 for Fortran. Unfortunately, we were not able to compare with other versions as we were limited to the modules available on Perlmutter.

5. I think hyper threading may be disabled in supercomputer. It would be helpful to omit the hyper-thread performance of the CPU in section 3.3.

   **Response:** We have removed the section on hyper-threading performance.

**Reviewer 1, Second Revision of Manuscript, "Julia for Geophysical Fluid Dynamics: Performance Comparisons between CPU, GPU, and Fortran-MPI"**

This is a revised submission. The authors have addressed most of my comments. I especially welcome the new "Julia in a Nutshell" section, which is overall very well done. I recommend acceptance after minor revisions.

**Minor Comments:**

1. Line 74: Julia's "superior memory management": I don't understand what the authors mean here. Julia is a garbage collected language which trades off programmer control for ease of memory management. Garbage collection can have undesirable effects in parallel applications (see for example this Julia issue https://github.com/JuliaLang/julia/issues/49316). It is not strictly superior to other memory management mechanisms.

   **Response:** We have removed "superior memory management" from this description.

2. Line 80: "Julia has found widespread application ...": I think this statement is too strong. Julia definitely does not have "widespread" application in web development, and my impression is that it is only gaining grounds in the other fields mentioned.

   **Response:** Agreed. We changed "found widespread application" to "recently been gaining ground".

3. Line 126: Strictly speaking Array is not a concrete type since it is parametric (this can be easily checked in Julia by evaluating isconcretetype(Array) which returns false). Additionally, here and in other places Julia types are written in plain text, but other times they are displayed as code. Please unify the style throughout the manuscript.

   **Response:** We have added specific text on how to make an Array concrete, which is by defining it with concrete-typed elements and specifying the Array's size. We have changed code within the text to code font.

4. There is a typo in equation (1a) (there should be only one gradient operator).

   **Response:** Thank you for pointing it out. We have removed the additional gradient operator.

5. I have a couple of suggestions for Table 1. Please mention that the TDP values are per CPU and per GPU. The A100 GPU comes in two variants with TDP of 300 W or 400 W. I believe that Perlmutter has the 400 W version. To be consistent with the GPU specification, please say that the CPU flops are for double precision.

   **Response:** Thank you for the suggestions. We have added these, and confirmed with NERSC that the TDP is 400W.

6. Lines 510-514: "Based on technical specification ...": The comparison of flops values implies that the authors think that their code is compute bound. Based on the low number of operations I would expect the code to be bandwidth limited and the bandwidth ratio to be a more appropriate speedup bound, at least for large problem sizes. Can the authors comment on this?

   **Response:** We added text to that paragraph to describe the timing of computation and communication separately. Based on Figure 5, the full-node Julia-MPI (64 cores) is 80% computation, so we do not believe the application is bandwidth limited on the 512x512 by 100 layer domain.

7. Line 618 "Julia-GPU scaled very similarly to Julia-MPI": Maybe "performed" would be better here than "scaled" ?

   **Response:** Agreed. The text has been updated.

8. Line 622: "(...) and sample results rarely, GPUs can offer significant speed-ups": This might be true in theory, but I don't see how the presented results support that conclusion. Looking at Figure 2, even

in the "Computation Only" plot the CPU is faster than the GPU, even though the GPU is theoretically much more powerful.

**Response:** Thank you for pointing this out. We have revised this paragraph to align with the results in the GPU/CPU comparison of the paper.